# DynaSchedBench: Calibrated Dynamic Scheduling Benchmarks and Observability Paradox in LLM-based Scheduling Agents

**Shijie Cao** [1 2]   **Yuan Yuan*** [1 3 4]   **Jing Liu** [5 6]

## Abstract

Progress in neural combinatorial optimization for Dynamic Flexible Job Shop Scheduling Problem (DFJSP) is currently hindered by a methodological tension: static benchmarks encourage benchmark overfitting, while uncalibrated generators obscure algorithmic capability with stochastic noise. To resolve this, we introduce **DynaSched-Bench**, a diagnostic framework for DFJSP that rigorously controls the instance-generation process. Instead of relying on parameter sampling, our approach utilizes Sequential Event-Space Calibrator (SESC) that computes a novel Schedule Stress Index (SSI) to stratify instances by difficulty. We demonstrate that SESC is substantially more computationally efficient than evolutionary baselines while converging reliably to the target metrics. The framework integrates modular components for instance generation, snapshot-based simulation, agents, evaluation, and visualization, thereby enabling rigorous testing of reactive and lookahead-based policies. Leveraging this calibrated environment, we identify key limitations of LLM-based scheduling agents. Specifically, in step-wise online decision-making for dynamic scheduling, we identify an "Observability Paradox": providing agents with oracle access to full structural information can degrade policy performance, underperforming concise information. Furthermore, despite substantial token overhead, tool-augmented and refinement strategies fail to reliably improve performance, and most LLM agents fail to consistently surpass strong dispatching baselines—behaving more like robust heuristic approximators than superior optimizers.

## 1. Introduction

The intersection of Operations Research (OR) and Artificial Intelligence has been reshaped by the rise of learning-based optimization methods. While classical exact methods and heuristics (Taillard, 1993; Demirkol et al., 1998) have long formed the foundation of production scheduling, they often struggle to respond to the rapid, stochastic event streams in Industry 4.0 settings (e.g., frequent job arrivals, machine breakdowns, and processing-time variability) (Ngwu et al., 2025). Consequently, Deep Reinforcement Learning (DRL) approaches, such as L2D (Zhang et al., 2020) and ScheduleNet (Park et al., 2021), have been proposed to learn dispatching policies that support real-time decision-making (Li et al., 2025). However, these models are prone to overfitting on fixed-size benchmark sets and offer limited semantic interpretability for human-in-the-loop systems. More recently, Large Language Models (LLMs) have emerged as a promising paradigm for reasoning-and-acting with tools (Yao et al., 2023b), offering a potential avenue to combine combinatorial structure with semantic interpretability (Fu et al., 2024).

Despite these algorithmic advances, further progress in Neural Combinatorial Optimization (NCO) is increasingly constrained by a methodological bottleneck in evaluation. Existing benchmarks, such as the widely used Taillard (Taillard, 1993) and DMU (Demirkol et al., 1998) instance sets, are static, finite, and deterministic. Training on these fixed sets inevitably leads to overfitting, where agents memorize specific instance structures rather than learning generalizable policies (Zhang & Zhu, 2025). This static evaluation paradigm misaligns with Dynamic Flexible Job Shop Scheduling (DFJSP), where the core challenge lies in managing continuous, stochastic dynamic events (Yang et al., 2025).

Furthermore, current attempts to generate dynamic scheduling instances typically rely on uncalibrated procedural sampling (Yu et al., 2026), where difficulty emerges as an uncontrolled consequence of random seeds. This lack of cal-

---
* Corresponding author. [1]School of Computer Science and Engineering, Beihang University, Beijing 100191, China [2]Shenzhen Loop Area Institute, Shenzhen, China [3]Qingdao Research Institute, Beihang University [4]Hangzhou Innovation Institute, Beihang University [5]School of Artificial Intelligence, Xidian University, Xi'an 710071, Shaanxi, China [6]Guangzhou Institute of Technology, Xidian University, Guangzhou 510555, Guangdong, China. Correspondence to: Yuan Yuan <yuan21@buaa.edu.cn>.

*Proceedings of the 43rd International Conference on Machine Learning*, Seoul, South Korea. PMLR 306, 2026. Copyright 2026 by the author(s).

ibration introduces a high-variance "stochastic fog" into evaluation: it becomes difficult to disentangle whether an algorithm's apparent gains stem from genuine innovation or simply from encountering a favorable sequence of dynamic events (Guillen-Perez et al., 2026). Without a principled mechanism to control instance difficulty and distribution, the field cannot reliably benchmark the reasoning capabilities of emerging LLM-based agents against established solvers.

To address this, we present **DynaSchedBench**, a principled framework for generating calibrated, theoretically grounded benchmark instances for DFJSP[1]. Our framework enables controlled stress-testing of scheduling agents, so that reported performance reflects robust adaptability rather than stochastic luck. Our contributions are as follows:

1. **Calibrated Generation via Event-Space Refinement:** We replace parameter sampling with direct event-stream transformations. By applying isomorphic resampling, our method aligns instances with target distributions significantly faster and more precisely than evolutionary baselines.

2. **Theoretical Difficulty Modeling via Schedule Stress Index (SSI):** We introduce SSI to quantify the interaction between variability and utilization. This metric enables the systematic stratification of instance difficulty—from "under-loaded" to "critical"—to rigorously map solver performance phase transitions.

3. **Modular Simulation and Evaluation Architecture:** We develop a Gym-compatible environment featuring efficient state snapshotting. This architecture supports lookahead-dependent reasoning strategies and employs a trajectory-based engine for reproducible constraint verification.

4. **Comprehensive Assessment of LLM Limitations:** We uncover an "Observability Paradox" where structural priors degrade LLM performance compared to statistical summaries. Furthermore, we demonstrate that expensive reasoning strategies yield diminishing returns, characterizing current LLMs as robust heuristic approximators rather than superior optimizers.

## 2. Related Work

**Job Shop Scheduling and Static Benchmarks.** The Job Shop Scheduling Problem (JSP) and its flexible variant are classical NP-hard problems (Cao & Yuan, 2025a). Classical benchmark sets by Taillard (Taillard, 1993) and Demirkol et al. (Demirkol et al., 1998) have shaped algorithm development for decades. These instances, however, are static and

deterministic. Therefore, they fail to capture key sources of stochasticity in real production systems, such as machine breakdowns, uncertain processing times, and random job arrivals.

**Deep Reinforcement Learning for Dynamic Scheduling.** In dynamic shop-floor environments, Priority Dispatching Rules (PDRs) are widely used due to their low computational cost and simple deployment (Corsini et al., 2024). However, designing effective PDRs typically requires substantial expert domain knowledge. This has motivated a surge of DRL methods that aim to automate policy design (Zhang et al., 2024; Xu et al., 2025). DRL-based schedulers include DAN (Wang et al., 2024b), DDQN (Zhang et al., 2026b), graph-attention architectures (Zhao et al., 2026), and PPO-based approaches (Yuan et al., 2025). Despite these advances, empirical studies report that DRL-based schedulers often generalize poorly to unseen instance sizes and distributions, a limitation highlighted in recent comprehensive surveys (Khadivi et al., 2025).

**LLMs for Combinatorial Optimization.** The emergence of LLMs has sparked interest in their potential as general tools for optimization and decision making. Approaches such as Chain-of-Thought (CoT) (Wei et al., 2022) and Tree of Thoughts (ToT) (Yao et al., 2023a) enable LLMs to decompose complex reasoning tasks. In optimization, LLMs have been explored as heuristic generators (Romera-Paredes et al., 2024), code-based solvers (Wang et al., 2024a; AhmadiTeshnizi et al., 2024), and direct decision-making agents (Abgaryan et al., 2024). However, recent studies suggest that LLMs can struggle with long-horizon planning and spatial reasoning, and may exhibit performance degradation when faced with long or overloaded contexts (Valmeekam et al., 2025). Our work examines these limitations in stepwise online decision making for dynamic scheduling, contributing to the growing literature on LLM reliability in combinatorial optimization (Wang et al., 2026).

## 3. Benchmark Instance Model and Calibrated Generation

Fig. 1 summarizes the overall framework of DynaSchedBench, including calibrated instance generation, SSI-based difficulty modeling, LLM-based scheduling agents, and the modular simulation–evaluation stack.

### 3.1. Input Configuration and Notation

A configurable benchmark generator for DFJSP is considered (Cao & Yuan, 2025b). An input configuration $\mathcal{I}$ parameterizes the generation process, including the plant structure, stochastic distributions, target metrics, and dynamic scenarios, and induces a distribution over instances and event

---

[1]Project Page at https://dsbx7.github.io/

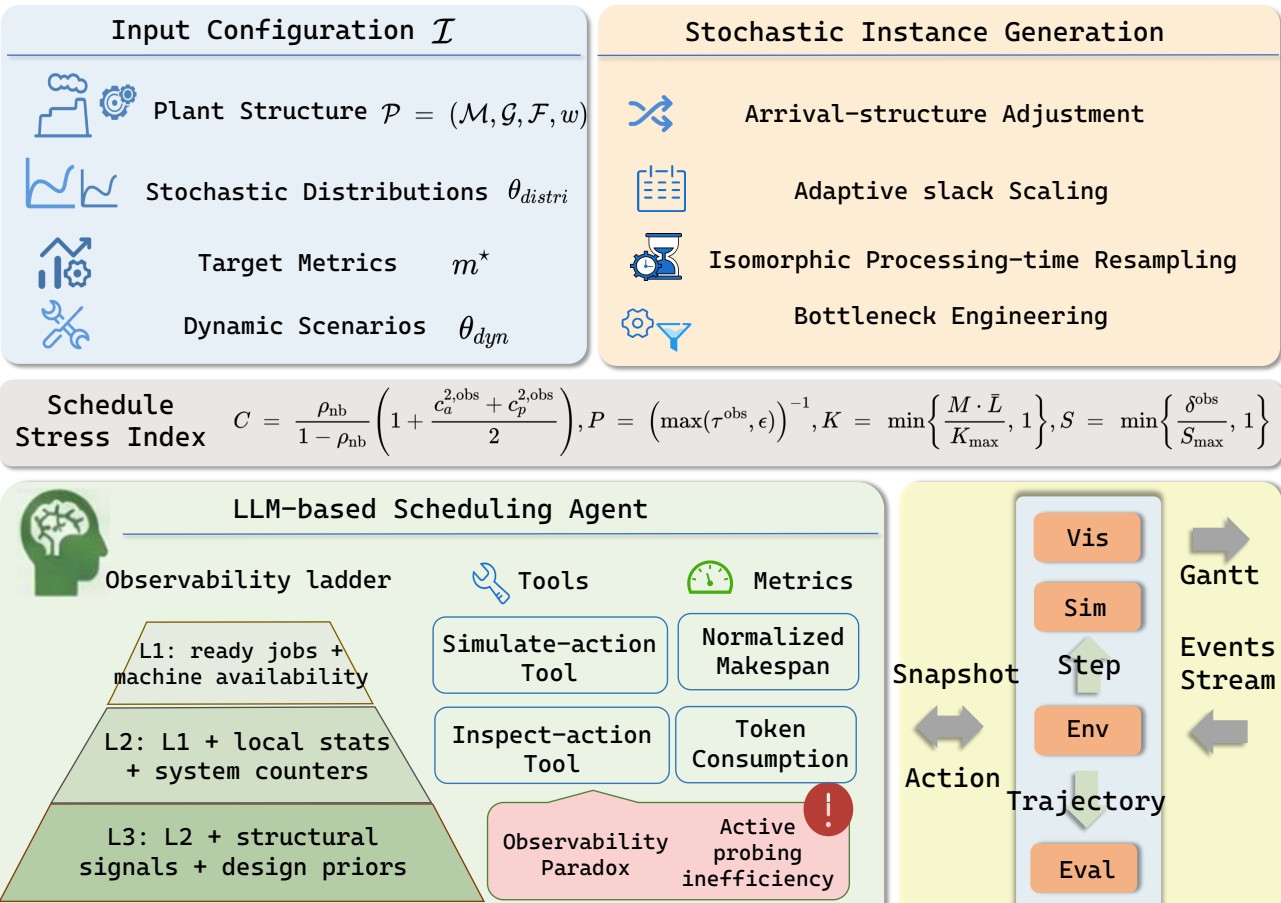

*Figure 1.* Overview of the DynaSchedBench framework. The system transforms input configurations into calibrated event streams using SESC and SSI. This rigorous environment supports the evaluation of LLM agents across different observability levels (L1–L3) and reasoning strategies.

streams. The configuration is organized as

$$\mathcal{I} = \big(\mathcal{P}, \theta_{\mathrm{distri}}, \boldsymbol{m}^\star, \theta_{\mathrm{dyn}}\big). \tag{1}$$

Superscript $^\star$ denotes a target metric value, and superscript $^{\mathrm{obs}}$ denotes the observed value computed from the realized event stream $\mathcal{E}$. A complete list of symbols and the configuration-layer specification are provided in Appendix A.

### 3.2. Stochastic Instance Generation

The core generation engine transforms the configuration $\mathcal{I}$ into a fully realized discrete event stream $\mathcal{E}$.

**Base Arrival Rate.** The base arrival rate $\lambda_{\mathrm{base}}$ is primarily determined by the target utilization $\rho_{\mathrm{global}}^\star$ via

$$\lambda_{\mathrm{base}} = \frac{\rho_{\mathrm{global}}^\star \sum_{m \in \mathcal{M}} v_m}{\mathrm{Mean}\, P}, \tag{2}$$

where $\mathrm{Mean}\, P$ is the expected work per job,

$$\mathrm{Mean}\, P = \mathbb{E}\left[\sum_{o \in \mathcal{O}_j} \mu_{f_j,o}\right] = \sum_{f \in \mathcal{F}} w_f \sum_{o \in \mathcal{K}_f} \mu_{f,o}, \tag{3}$$

If a fixed job count $N_J^{\mathrm{fix}}$ is specified, then

$$\lambda_{\mathrm{base}} = \frac{N_J^{\mathrm{fix}}}{H}. \tag{4}$$

where $H$ denotes the simulation horizon.

**Stationary Renewal Arrivals.** A stationary renewal arrival process is generated by sampling i.i.d. inter-arrival times from a Gamma distribution $\Gamma(k_a, \theta_a)$ (Lassoued et al., 2026). The shape and scale parameters are chosen to match the prescribed arrival-rate $\lambda_{\mathrm{base}}$ and squared coefficient of variation $c_a^{2\star}$:

$$k_a = (c_a^{2\star})^{-1}, \quad \theta_a = (k_a \lambda_{\mathrm{base}})^{-1}. \tag{5}$$

**Nonstationary Arrivals via Time Warping.** To represent nonstationary arrivals, the generator specifies a time-varying intensity function $\lambda(t)$ through a deterministic modulation envelope. Let $A \in [0, 1]$ denote the modulation amplitude, $T$ the modulation period. We write $\lambda(t) = \lambda_{\text{base}} g(t)$, where the supported modulation profiles are

$$g(t) = \begin{cases} 1, & \text{constant,} \\ 1 + A\sin\left(\dfrac{2\pi t}{T}\right), & \text{periodic,} \\ 1 + A\left(\dfrac{2t}{H} - 1\right), & \text{linear trend.} \end{cases} \quad (6)$$

Given baseline renewal timestamps $T_k^{\text{raw}}$ generated under the stationary rate $\lambda_{\text{base}}$, we define the cumulative modulation map

$$\Lambda(t) = \int_0^t g(s)\, ds. \quad (7)$$

The warped timestamp is then obtained by the monotone time change

$$T_k = \Phi(T_k^{\text{raw}}), \qquad \Phi = \Lambda^{-1}. \quad (8)$$

Since $g(t) \geq 0$ for the supported profiles, $\Lambda(t)$ is nondecreasing, so the transformation preserves the arrival order while redistributing events toward high-intensity periods.

**Base Due-Date.** Baseline due dates are set using a due-date tightness parameter $\tau^\star$. For a job $j$ with arrival time $t_j$ and workload $W_j$, the baseline due date is defined as (Liu et al., 2022)

$$D_j^{(0)} = t_j + \tau^\star W_j, \quad (9)$$

and is truncated to satisfy feasibility within the effective horizon. Due-date update events are specified by the dynamic-scenario layer, so $D_j^{(0)}$ serves as a baseline due date that may be modified over time by subsequent due-date-change events.

**Disturbances and Exogenous Events.** Capacity loss is realized through exogenous outages so that the total lost capacity matches the prescribed disturbance ratio $\delta^\star$ (Pal et al., 2023). Let the nominal total capacity over the horizon $H$ be

$$C_{\text{tot}} = H \sum_{m \in \mathcal{M}} v_m, \quad (10)$$

and define the corresponding global loss budget as

$$\Delta C = \delta^\star C_{\text{tot}}. \quad (11)$$

For each bottleneck window $b \in \mathcal{B}$ associated with group $g_b$ and time interval $[s_b, e_b]$, let $H_b = e_b - s_b$ and

$$C_b = H_b \sum_{m \in g_b} v_m, \qquad \Delta C_b = \left[ C_b - \frac{W_b}{\rho_b^\star} \right]_+, \quad (12)$$

where $W_b$ denotes the nominal workload released to group $g_b$ within $[s_b, e_b]$ and $[x]_+ = \max\{x, 0\}$. Outage events from breakdowns, repairs, and preventive maintenance are scheduled on machines in $g_b$ to realize the budgets $\Delta C$ and $\{\Delta C_b\}_{b \in \mathcal{B}}$ (Wocker et al., 2024).

**Routing, Processing Times, and Batch Arrivals.** For each job $j$, a process template $f_j$ is sampled from the mixture $w$, which determines the operation sequence.

Processing times are then sampled from a Gamma distribution parameterized by the nominal mean $\mu_{f_j,o}$ and the target processing-time variability $c_p^{2\star}$.

Batch sizes $B$ are sampled from a normal distribution $\widetilde{B} \sim \mathcal{N}(\mu_B, \sigma_B^2)$, generating $B = \max(1, \lceil \widetilde{B} \rceil)$ jobs that share the same arrival time and template (Zhu et al., 2026).

### 3.3. Metrics and Difficulty Modeling

The metrics engine performs post-hoc diagnostics on the realized discrete-event stream $\mathcal{E}$, producing the empirical metric vector $m^{\text{obs}}$ and a composite difficulty score via the SSI.

**Empirical moments and workload accounting.** Let $\mathcal{E}_{\text{arr}}$ denote the set of job-arrival events. We index the realized jobs by $\mathcal{J} = \{1, \ldots, N_J\}$ with $N_J = |\mathcal{E}_{\text{arr}}|$, and denote the nondecreasing arrival timestamps by $\{t_j\}_{j \in \mathcal{J}}$. Inter-arrival times are $\Delta t_j = t_j - t_{j-1}$ for $j = 2, \ldots, N_J$. Let $\mathcal{P}_{\text{exec}}$ be the multiset of realized operation processing times extracted from $\mathcal{E}$, and denote its elements by $p$. The engine computes empirical means and variances $\mu_{\Delta t}, \sigma_{\Delta t}^2$ from $\{\Delta t_j\}_{j=2}^{N_J}$ and $\mu_p, \sigma_p^2$ from $\mathcal{P}_{\text{exec}}$.

**Global utilization and load imbalance.** Let $W_g$ be the realized workload processed by machine group $g \in \mathcal{G}$, defined as the sum of realized processing durations of operations executed on machines in $g$, and define $W_{\text{tot}} = \sum_{g \in \mathcal{G}} W_g$. The observed global utilization is

$$\rho_{\text{global}}^{\text{obs}} = \frac{W_{\text{tot}}}{C_{\text{tot}}} = \frac{\sum_{g \in \mathcal{G}} W_g}{H \sum_{m \in \mathcal{M}} v_m}. \quad (13)$$

For each group $g \in \mathcal{G}$, let $C_g = H \sum_{m \in g} v_m$ and $\rho_g^{\text{obs}} = W_g / C_g$. We quantify load imbalance via the coefficient of variation of group utilizations:

$$\chi_{\text{load}}^{\text{obs}} = \frac{\sqrt{\frac{1}{|\mathcal{G}|} \sum_{g \in \mathcal{G}} \left( \rho_g^{\text{obs}} - \text{Mean } \rho \right)^2}}{\text{Mean } \rho}. \quad (14)$$

**Time-windowed bottleneck diagnostics.** Each bottleneck specification $b \in \mathcal{B}$ is associated with a time window $[s_b, e_b]$, a machine group $g_b \in \mathcal{G}$, and a target utilization

$\rho_b^\star$. Let $\tilde{\mathcal{D}}_m$ be the set of disjoint effective downtime intervals for machine $m$, obtained by unioning and merging raw downtime intervals extracted from $\mathcal{E}$. Define the downtime duration of machine $m$ within window $b$ as

$$T_{m,b}^{\text{down}} = \sum_{[\tilde{s},\tilde{e}] \in \tilde{\mathcal{D}}_m} \left|[\tilde{s},\tilde{e}] \cap [s_b, e_b]\right|, \qquad (15)$$

where $|\cdot|$ denotes interval length. The effective capacity in window $b$ is then

$$C_b^{\text{eff}} = \sum_{m \in g_b} v_m\Big((e_b - s_b) - T_{m,b}^{\text{down}}\Big). \qquad (16)$$

Let $W_b^{\text{win}}$ be the realized workload executed by group $g_b$ within $[s_b, e_b]$. The observed bottleneck utilization is

$$\rho_b^{\text{obs}} = \frac{W_b^{\text{win}}}{C_b^{\text{eff}}} \qquad (17)$$

**Variability and due-date tightness.** The realized variability is measured by SCV,

$$c_a^{2,\text{obs}} = \left(\frac{\sigma_{\Delta t}}{\mu_{\Delta t}}\right)^2, \qquad c_p^{2,\text{obs}} = \left(\frac{\sigma_p}{\mu_p}\right)^2. \qquad (18)$$

Let $D_j$ denote the final due date of job $j$ after any due-date update events, and let $W_j$ be the realized total processing workload of job $j$. We define the normalized slack factor

$$\tau^{\text{obs}} = \frac{1}{N_J} \sum_{j \in \mathcal{J}} \left(\frac{D_j - t_j}{W_j}\right), \qquad (19)$$

and use $\tau^{\text{obs}}$ as the empirical due-date tightness indicator.

**Observed disturbance.** For each machine $m$, we extract raw unavailability intervals and take their union to obtain disjoint effective intervals $\tilde{\mathcal{D}}_m$. The total observed lost capacity is

$$C_{\text{down}}^{\text{obs}} = \sum_{m \in \mathcal{M}} v_m \sum_{[\tilde{s},\tilde{e}] \in \tilde{\mathcal{D}}_m} (\tilde{e} - \tilde{s}), \qquad (20)$$

and the realized disturbance ratio is

$$\delta^{\text{obs}} = \frac{C_{\text{down}}^{\text{obs}}}{C_{\text{tot}}}. \qquad (21)$$

### 3.4. Schedule Stress Index

SSI maps raw metrics into a four-component difficulty vector capturing congestion, time pressure, structural complexity, and disruption intensity.

**Congestion component** ($C$). Motivated by Kingman-style heavy-traffic approximations (Kingman, 1961), we use the natural bottleneck utilization $\rho_{\text{nb}} = \max_{g \in \mathcal{G}} \rho_g^{\text{obs}}$ and define

$$C = \frac{\rho_{\text{nb}}}{1 - \rho_{\text{nb}}}\left(1 + \frac{c_a^{2,\text{obs}} + c_p^{2,\text{obs}}}{2}\right). \qquad (22)$$

**Time-pressure component** ($P$). We define time pressure as the inverse of empirical slack (Yoo et al., 2025), using a small constant $\epsilon > 0$ for numerical stability:

$$P = \Big(\max(\tau^{\text{obs}}, \epsilon)\Big)^{-1}. \qquad (23)$$

**Structural-complexity component** ($K$). Let $M = |\mathcal{M}|$ and let $L_j$ be the realized number of operations of job $j$, with $L_j = |\mathcal{K}_{f_j}|$ under fixed templates. Define $\text{Mean } L = \frac{1}{N_J} \sum_{j \in \mathcal{J}} L_j$ and a user-specified normalization constant $K_{\max} > 0$. We define the normalized structural complexity as

$$K = \min\left\{\frac{M \cdot \text{Mean } L}{K_{\max}}, 1\right\}. \qquad (24)$$

**Disruption-intensity component** ($S$). Using the realized disturbance ratio, we define a normalized disruption level

$$S = \min\left\{\frac{\delta^{\text{obs}}}{S_{\max}}, 1\right\}, \qquad (25)$$

where $S_{\max} > 0$ is a user-specified normalization constant corresponding to the maximum disturbance level represented on the SSI scale.

**Scalar difficulty score.** Let $\ell(x) = \log(1 + x)$ be a log-compression map. With component-wise normalizers $(C_{\max}, P_{\max}, K_{\max}, S_{\max})$, we compute

$$\hat{X} = \frac{\ell(X)}{\ell(X_{\max})} \quad \text{for } X \in \{C, P, K, S\}. \qquad (26)$$

We then define the scalar difficulty score as

$$d = 100 \cdot \frac{1}{4}\left(\hat{C} + \hat{P} + \hat{K} + \hat{S}\right), \qquad (27)$$

yielding a scalar difficulty score $d \in [0, 100]$. SSI rankings remain stable under component-weight perturbations, internal normalization sweeps, and alternative downstream normalizers. Appendix I reports these robustness checks.

### 3.5. Calibration Methods

To bridge the gap between stochastic instance generation and prescribed target requirements, we use Sequential Event-Space Calibrator (SESC) that operates directly on the realized discrete-event stream.

**Metric error objective.** Let

$$\mathcal{L} = \{\rho_{\text{global}}, c_a^2, c_p^2, \tau, \chi_{\text{load}}, \delta, \varepsilon_{\text{bn}}\} \qquad (28)$$

be the index set of calibrated metrics defined in Section 3.3. For each $\ell \in \mathcal{L}$, let $m_\ell^\star$ and $m_\ell^{\text{obs}}$ denote the target and observed values, respectively.

We define a normalized scalar error

$$e_\ell(m_\ell^{\mathrm{obs}}, m_\ell^\star) = \begin{cases} \dfrac{|m_\ell^{\mathrm{obs}} - m_\ell^\star|}{|m_\ell^\star| + \varepsilon}, & \text{if } |m_\ell^\star| > 0, \\ |m_\ell^{\mathrm{obs}} - m_\ell^\star|, & \text{if } |m_\ell^\star| = 0, \end{cases} \quad (29)$$

where $\varepsilon > 0$ is a small regularization constant. For the bottleneck component we set $m_{\varepsilon_{\mathrm{bn}}}^{\mathrm{obs}} = \varepsilon_{\mathrm{bn}}$ and $m_{\varepsilon_{\mathrm{bn}}}^\star = 0$, so that $e_{\varepsilon_{\mathrm{bn}}} = \varepsilon_{\mathrm{bn}}$ corresponds to the root-mean-square deviation across all bottleneck windows. Collecting all components yields the error vector $\boldsymbol{e} = (e_\ell)_{\ell \in \mathcal{L}}$.

**Sequential Event-space Calibrator.** Due to the strong coupling among utilization, variability, due-date tightness, disturbance, and bottleneck behavior, we adopt an event-space calibrator that applies a sequence of local structural transformations directly to the instantiated event list $\mathcal{E}$. As a baseline for comparison, we also consider parameter-space multi-objective optimization (MOO) and hybrid approaches (Hybrid) and the detailed design of calibrators is provided in Appendix B.

Let $\mathcal{S}$ denote a catalog of adjustment strategies. Each strategy $s \in \mathcal{S}$ defines an operator

$$\mathcal{T}_s : \mathcal{E} \longmapsto \mathcal{E}', \quad (30)$$

which modifies arrivals, processing times, due dates, or downtime events in a controlled way. The calibration loop performs a greedy search over $\mathcal{S}$, repeatedly selecting the operator that most reduces the error vector $\boldsymbol{e}$ while preserving metrics that are already close to their targets.

**Strategy selection policy.** For each strategy $s \in \mathcal{S}$, we specify a static impact vector $\boldsymbol{a}_s \in [-1, 1]^{|\mathcal{L}|}$. The component $a_{s,\ell}$ encodes the expected signed sensitivity of metric $\ell$ to strategy $s$, where positive values indicate improvement toward the target and negative values indicate deterioration. Given the current normalized error state $\boldsymbol{e}$, we define the utility of strategy $s$ as

$$\mathrm{Score}(s) = \sum_{\ell \in \mathcal{L}} \varphi(a_{s,\ell}, e_\ell). \quad (31)$$

where the weighting function $\varphi$ introduces an asymmetric penalty structure:

$$\varphi(a, e) = \begin{cases} a\,e, & a > 0, \\ -\lambda_{\mathrm{soft}}\,e\,|a|, & a < 0 \text{ and } e < \varepsilon_{\mathrm{tol}}, \\ -\lambda_{\mathrm{hard}}\,e\,|a|, & a < 0 \text{ and } e \geq \varepsilon_{\mathrm{tol}}. \end{cases} \quad (32)$$

Here $\varepsilon_{\mathrm{tol}} > 0$ is a convergence tolerance, typically $\varepsilon_{\mathrm{tol}} \approx 0.05$, and the penalty coefficients satisfy $\lambda_{\mathrm{hard}} \gg \lambda_{\mathrm{soft}}$. Intuitively, the first case rewards strategies that reduce metrics with large residual error, while the second and third cases downweight strategies that degrade any metric, with a stronger penalty when that metric has not yet converged.

**Adjustment strategy catalog.** The framework includes a set of specialized strategies, each targeting a subset of the metrics in $\mathcal{L}$:

1. **Arrival-structure adjustment.** This strategy primarily affects $\rho_{\mathrm{global}}$ and $c_a^2$ via two operations.

   Rate correction randomly deletes jobs and their associated events or duplicates existing jobs to adjust the effective total workload $\sum_{j \in \mathcal{J}} W_j$ when utilization errors are significant.

   Renewal resampling preserves the job order but regenerates the arrival timestamp vector $(t_j)_{j=1}^{N_J}$ from a Gamma distribution $\Gamma(k_a', \theta_a')$ chosen to match the target SCV $c_a^{2\star}$, followed by a deterministic time-warping step to fit the horizon $H$ without altering the total workload.

2. **Adaptive slack scaling.** This strategy targets the due-date tightness metric $\tau$ by manipulating the realized due-date slack $\xi_j = D_j - t_j$. We apply a multiplicative scaling factor $\alpha > 0$ to obtain updated due dates

$$D_j' = t_j + \mathrm{clip}\big(\alpha\,\xi_j;\, \xi_j^{\min},\, H - t_j\big), \quad (33)$$

   where $\mathrm{clip}(x; a, b) = \min\{\max\{x, a\}, b\}$ enforces feasibility within the planning horizon, and $\xi_j^{\min}$ is a lower bound on admissible slack. The scaling factor is updated iteratively according to

$$\alpha_{t+1} = \alpha_t\big(1 - \eta_\tau \, \mathrm{sgn}(\tau^{\mathrm{obs}} - \tau^\star)\, e_\tau\big), \quad (34)$$

   where $\eta_\tau > 0$ is a step-size parameter and $e_\tau$ is the current normalized error for $\tau$.

3. **Isomorphic processing-time resampling.** This strategy corrects $c_p^2$ while preserving global utilization. For all realized operations, we first draw new processing durations $\tilde{p}_{j,o}$ from a Gamma distribution whose parameters match the target processing-time variability $c_p^{2\star}$. To maintain the total workload, we then rescale these samples via

$$p_{j,o}' = \tilde{p}_{j,o} \cdot \frac{\sum_{j \in \mathcal{J}} \sum_{o \in \mathcal{K}_{f_j}} p_{j,o}}{\sum_{j \in \mathcal{J}} \sum_{o \in \mathcal{K}_{f_j}} \tilde{p}_{j,o}}, \quad (35)$$

   which preserves $\sum_{j,o} p_{j,o}$ exactly while adjusting the empirical SCV.

4. **Bottleneck engineering.** This strategy provides fine-grained control over bottleneck behavior, targeting the aggregate deviation $\varepsilon_{\mathrm{bn}}$. For a bottleneck window $b$ with utilization deviation $\Delta\rho_b = \rho_b^{\mathrm{obs}} - \rho_b^\star$, we compute the required adjustment in available capacity $\Delta C_b'$ from the bottleneck model. Let $v_{g_b} = \sum_{m \in g_b} v_m$ be the aggregate processing rate of group $g_b$, and let

$\mathcal{E}_{\text{out}} \cap b$ denote unavailability events overlapping the window. The strategy perturbs the durations of these events or inserts compensating repair or maintenance events so that

$$\sum_{e \in \mathcal{E}_{\text{out}} \cap b} \text{dur}(e)' \approx \sum_{e \in \mathcal{E}_{\text{out}} \cap b} \text{dur}(e) + \frac{\Delta C_b'}{v_{g_b}}. \quad (36)$$

## 4. DynaSchedBench Architecture and Stack

The architecture is organized into six decoupled subsystems: Generation, Simulation, Environment, Agents, Evaluation, and Visualization.

### 4.1. Instance Generation and Simulation Kernel

The Generation module (Gen) transforms the high-level input model described in Section 3.1 into calibrated event streams. The Simulation module (Sim) implements a discrete-event engine. To support diverse agent implementations, including memoryless LLM-based policies and lookahead planners, the simulator encapsulates the mutable environment state and only exposes immutable observations. At each decision epoch, it exports a snapshot of the system, ensuring that agent decisions cannot corrupt the simulation clock or state consistency.

### 4.2. Standardized Environment and Agent Interface

The Environment module (Env) wraps the simulation kernel into a standardized Gym-like interface with reset and step functions (Towers et al., 2024). It translates raw snapshots into structured observations such as ready operations and machine availability, and filters admissible actions to ensure validity. The Agents module (Agent) defines a unified protocol for solvers and abstracts differences among PDRs, RL, and LLMs.

### 4.3. Trajectory-Based Evaluation and Visualization

To enable rigorous post-hoc analysis, the framework employs a streaming Evaluation system (Eval). It persists execution traces as structured trajectories. These trajectories are consumed by the Visualization module (Vis) to generate Gantt charts, event timelines, and stability metric curves, and by the diagnostic engine to verify hard constraints such as precedence and resource non-overlapping and to compute performance metrics (Cao et al., 2023).

## 5. LLM-based Scheduling Agents

We define three observability levels: L1 (local view), L2 (adds statistics), and L3 (adds structural priors). Agents use two tools: a *simulation* tool providing completion-time proxies and an *inspection* tool for feature descriptions.

Based on $O_t$ and $\mathcal{A}_t$, agents use direct (Zhang et al., 2026a), CoT (Wei et al., 2022), or tool-augmented (Wang et al., 2025) prompting. Optional refinement strategies include greedy selection, reflection (Shinn et al., 2023), and best-of-$n$.

## 6. Calibrator Experiments

### 6.1. Experimental Configuration

Evaluations are performed on the **DynaSched-Grid** and **DynaSched-Sweep** benchmarks to validate the proposed SESC against baseline methods, including MOO and Hybrid Calibrator. Detailed dataset specifications and the configurations for the scaling study are provided in Appendix C.

For each calibration task, the objective is to minimize the normalized distance between the observed metric vector $m^{\text{obs}}$ and the target vector $m^\star$. Performance is quantified through the calibration loss $\ell_2 = \|m^{\text{obs}} - m^\star\|_2$, the strict success rate defined by the simultaneous satisfaction of all metric-wise tolerances, and the computational overhead measured by wall-clock execution time.

### 6.2. Calibration Performance Analysis

To establish stable operating points, hyperparameters are selected using three robustness filters: Accuracy ($Acc$), 90th percentile $L_2$ error ($P90$), and scenario coverage ($Cov$). Defaults were chosen to maximize coverage subject to accuracy constraints: SESC is fixed at $Cov = 1.0$ with $Acc \geq 0.9$, while MOO and Hybrid are set to $Cov \geq 0.65$ with $Acc \geq 0.8$. These standardized configurations—labeled Best SESC, Best MOO, and Best Hybrid—are used for all subsequent experiments. Sensitivity analyses are detailed in Appendix D. An operator ablation shows that due-date slack scaling is the dominant convergence driver and that the full operator set is needed for tail robustness and details are reported in Appendix G.5.

### 6.3. Stochastic Stability Analysis

The robustness of the calibration process is evaluated by repeating the procedure across multiple random seeds for each scenario. In Table 1, the SESC demonstrates superior algorithmic stability compared to the MOO and Hybrid baselines, characterized by significantly lower mean normalized loss $\tilde{\ell}_2$ and minimal seed-to-seed dispersion. This stability confirms that the proposed method is less sensitive to the initial stochastic state of the event stream. Detailed statistical summaries and per-scenario case studies are provided in Appendix F.

*Table 1.* Calibration accuracy, seed-level variability, and computational overhead across operational regimes.

| Mode | $L_2$ | | | Runtime (s) | | | Mean $L_2$ across seeds | | | Std $L_2$ across seeds | | |
|---|---|---|---|---|---|---|---|---|---|---|---|---|
| | Median | $P_{25}$ | $P_{75}$ | Median | $P_{25}$ | $P_{75}$ | Median | $P_{25}$ | $P_{75}$ | Median | $P_{25}$ | $P_{75}$ |
| SESC | 0.0539 | 0.0362 | 0.0829 | 0.2 | 0.1 | 1.1 | 0.0524 | 0.0508 | 0.0602 | 0.0236 | 0.0203 | 0.0352 |
| MOO | 0.0826 | 0.0491 | 0.1719 | 57.9 | 12.4 | 181.3 | 0.1049 | 0.0966 | 0.2217 | 0.1532 | 0.1068 | 0.1804 |
| Hybrid | 0.0595 | 0.0329 | 0.1082 | 185.8 | 40.4 | 611.5 | 0.0718 | 0.0458 | 0.1166 | 0.0562 | 0.0280 | 0.0812 |

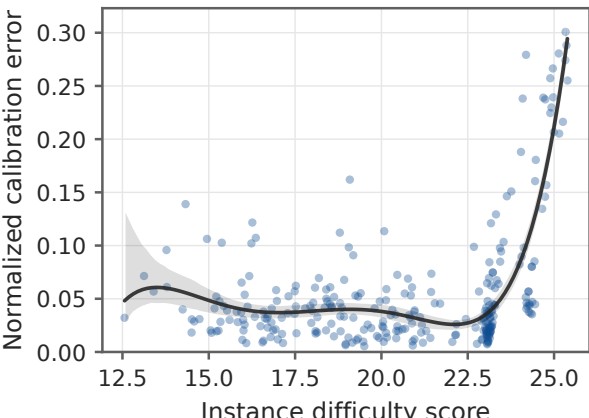

*Figure 2.* Calibration error as a function of instance difficulty score. The cubic fit indicates an accelerating difficulty curve.

### 6.4. Difficulty Score Validation

Fig. 2 compares the proposed SSI with the observed calibration error. Overall, the scatter shows an approximately monotonic pattern in which higher difficulty scores are associated with larger residual calibration errors. The fitted trend indicates that errors are relatively stable over the mid-range and rise sharply near the upper end of the difficulty scale.

We further evaluate SSI against an outcome-side difficulty measure, defined as the makespan gap between a weak random scheduler and the best fixed heuristic, $\Delta_{\text{gap}} = C_{\max}(\text{random}) - C_{\max}(\text{best})$. On the DynaSched-Subset, SSI is positively correlated with this scheduling gap (Spearman $\rho = 0.563$, $p = 3.85 \times 10^{-7}$). Moreover, the top SSI quintile has a substantially larger mean gap than the bottom quintile (694.12 vs. 146.71), indicating that high-SSI instances are not only harder to calibrate but also harder to schedule. Detailed quantile statistics are reported in Table 2.

## 7. LLM-Based Scheduling Experiments

### 7.1. Experimental Configuration

To ensure statistical reliability under computational constraints, we construct **DynaSched-Subset** using farthest-point $k$-center sampling over the driver-metric manifold. The subset contains 70 instances, with 30 from DynaSched-

*Table 2.* Outcome-side validation of SSI using scheduling difficulty. The first two rows report mean scheduling gaps for the bottom and top SSI quintiles. The difference reports the observed mean difference, with the confidence interval estimated by bootstrap resampling.

| SSI group | Mean gap | 95% bootstrap CI |
|---|---|---|
| Bottom 20% | 146.71 | [57.41, 289.31] |
| Top 20% | 694.12 | [479.22, 914.13] |
| Difference | +547.41 | [283.47, 804.00] |

*Table 3.* Cost-performance metrics across LLM-based Scheduling configurations.

| Configuration | Tokens ($10^6$) | Mean $C_{max}(\%)$ | SE(%) |
|---|---|---|---|
| L1 Direct | 3.752 | 2.0 | 1.0157 |
| L1 CoT | 4.016 | 1.9 | 1.0151 |
| L1+Tool | 12.614 | 2.0 | 1.0271 |
| L2 CoT | 6.883 | 0.7 | 0.1375 |
| L2 Reflection | 14.470 | 1.7 | 1.0218 |
| L2 BestOfN | 7.770 | 2.7 | 1.5060 |
| L3 CoT | 8.620 | 1.7 | 1.0173 |

Grid and 40 from DynaSched-Sweep. It preserves the full-set interquartile range. The Kolmogorov–Smirnov (KS) statistics are 0.175 for difficulty, 0.086 for utilization, and 0.251 for arrival SCV. Appendix H reports the subset selection and coverage diagnostics.

The observability comparison in Table 3 uses the specified L1, L2, and L3 observation encodings for each configuration. The cross-model comparison in Table 5 uses a matched L1 Direct protocol with the same prompt templates, two few-shot examples, temperature 0.0, and top-$p = 1.0$.

Inference is performed using a suite of LLMs including Gemini (Gemini Team, 2025), DeepSeek (DeepSeek-AI, 2025), Kimi (Kimi Team, 2025), Grok, Qwen (Qwen Team, 2025), Claude, and GPT families. Open-weight models were served locally using the vLLM engine (Kwon et al., 2023) on a node with Intel Xeon Platinum 8468V CPUs and NVIDIA H100 80GB GPUs. Performance is quantified through normalized makespan $C_{\max}$ relative to the best observed schedule on each instance and total token consumption to characterize the cost-performance trade-off.

## 7.2. The Observability Paradox and Efficiency Bottlenecks

Tables 3 and 4 uncover an "Observability Paradox": providing full structural priors (L3) degrades performance compared to concise summaries (L2) (1.66% vs. 0.65%), implying that current LLMs struggle to distill useful signals from high-dimensional noise. Additionally, tool-augmented exploration incurs a three-fold token cost increase over L1 CoT while performing worse, highlighting severe efficiency bottlenecks in iterative reasoning.

*Table 4.* Distributional relative-to-best makespan gap for observability variants.

| Configuration | Mean | p50 | p90 | p95 |
|---|---|---|---|---|
| L1 Direct | 1.95 | 0.63 | 2.57 | 3.27 |
| L1+Tool | 2.03 | 0.52 | 2.47 | 4.27 |
| L2 CoT | 0.65 | 0.25 | 1.70 | 2.01 |
| L3 CoT | 1.66 | 0.35 | 1.95 | 2.92 |

The distributional view in Table 4 shows that the L2 advantage is not only an average effect. L2 CoT also has a lower p95 tail than L3 CoT (2.01% vs. 2.92%), indicating greater tail stability under concise statistical observations.

## 7.3. Performance Bounds and Heuristic Parity

*Table 5.* Comparison of LLM-based scheduling and priority dispatching rules. The results highlight that LLMs currently approximate rather than exceed the performance of optimized hand-crafted heuristics.

| Type | Method | Mean $C_{\max}$ (%) | SE (%) |
|---|---|---|---|
| LLM | Qwen3-4B | 1.45 | 0.2163 |
| | Qwen3-8B | 1.01 | 0.2173 |
| | Qwen3-14B | 1.34 | 0.1955 |
| | Qwen3-32B | 1.65 | 0.2072 |
| | Kimi-K2 | 1.46 | 0.2383 |
| | Claude-4.5 Haiku | 1.52 | 0.2342 |
| | DeepSeek-V3.2 | 1.68 | 0.2943 |
| | Gemini-2.5 FlashLite | 1.76 | 0.2921 |
| | Grok-4 Fast | 1.88 | 0.2316 |
| | GPT-5 Mini | 1.93 | 0.2469 |
| Heuristic | Best heuristic | 1.11 | 0.1540 |
| | Median heuristic | 1.94 | 0.2479 |
| | Worst heuristic | 51.52 | 3.8762 |

The heuristic pool contains 24 composite PDRs formed by 8 sequencing rules (SPT, LPT, FIFO, LIFO, LWKR, MWKR, LOPNR, MOPNR) and 3 machine-assignment rules (LIT, LWL, SPT). The best heuristic is the single fixed rule, not a per-instance oracle.

Table 5 reveals that LLMs congregate in a narrow performance band (Mean $C_{\max} \approx 1.01\%$–$1.93\%$), matching strong heuristics without breaching the performance ceil-

ing. While avoiding the catastrophic failures of weak PDRs, LLMs act primarily as robust "safe" approximators rather than transformative optimizers. Paired tests reinforce this heuristic-ceiling interpretation. Qwen3-8B is statistically indistinguishable from the strongest rule LIFO+LIT. The same test cleanly separates Qwen3-8B and Claude-4.5 Haiku from a weak SPT+SPT rule. Table 6 gives the paired-test details.

*Table 6.* Paired relative-best significance tests for the heuristic-ceiling claim. Negative $\Delta$ means Method A has a lower relative-best gap.

| Method A | Method B | $\Delta$ (pp) | 95% CI | FDR $p$ |
|---|---|---|---|---|
| Qwen3-8B | LIFO+LIT | -0.099 | [-0.531, 0.389] | 0.397 |
| Qwen3-8B | SPT+SPT | -50.510 | [-57.721, -43.246] | < 0.001 |
| Claude Haiku | SPT+SPT | -49.997 | [-57.401, -42.612] | < 0.001 |

The inability of top models like Claude to consistently outperform PDRs suggests limited combinatorial intuition, highlighting the need to pivot research from reactive dispatching to multi-step trajectory planning.

## 8. Conclusion

In this work, we address the evaluation crisis in dynamic scheduling by introducing DynaSchedBench, a principled framework for generating calibrated, distributionally controlled benchmarks. Our extensive evaluation of LLM-based agents, enabled by the framework's modular architecture, uncovers a critical "Observability Paradox": current models struggle to distill effective strategies from high-dimensional structural priors and remain bounded by the performance of classical heuristics.

## Acknowledgments

This work was partially supported by Shenzhen Loop Area Institute under Project No. FP202602. The authors gratefully acknowledge the support from the project team members involved in this work.

## Impact Statement

This paper presents work whose goal is to advance the field of Machine Learning. There are many potential societal consequences of our work, none which we feel must be specifically highlighted here.

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

# A. Notation and Configuration-Layer Specification

This appendix provides a comprehensive reference for the mathematical notation and configuration parameters used in the paper.

## A.1. Input Configuration and Plant Structure

The benchmark configuration is defined as a tuple $\mathcal{I} = (\mathcal{P}, \theta_{\text{distri}}, \boldsymbol{m}^{\star}, \theta_{\text{dyn}})$. Table a-1 covers the plant structure and input configuration.

*Table a-1.* Plant Structure and General Notation

| Symbol | Description |
|---|---|
| $\mathcal{I}$ | Input configuration tuple |
| $\mathcal{P}$ | Plant structure specification $(\mathcal{M}, \mathcal{G}, \mathcal{F}, \boldsymbol{w})$ |
| $\mathcal{M}$ | Finite set of machines |
| $m$ | Machine index, $m \in \mathcal{M}$ |
| $v_m$ | Speed factor of machine $m$ |
| $\mathcal{G}$ | Partition of machines into groups |
| $g$ | Machine group index, $g \in \mathcal{G}$ |
| $\mathcal{F}$ | Library of process templates |
| $f$ | Process template index, $f \in \mathcal{F}$ |
| $\mathcal{K}_f$ | Ordered set of operations for template $f$ |
| $o$ | Operation index |
| $\mathcal{O}_j$ | Ordered set of operations belonging to job $j$ |
| $\mu_{f,o}$ | Nominal mean processing time for operation $o$ in template $f$ |
| $\boldsymbol{w}$ | Job-mix probability vector ($\sum w_f = 1$) |
| $H$ | Simulation horizon |
| $N_J^{\text{fix}}$ | Fixed job count (optional constraint) |
| $\mathcal{J}$ | Set of realized jobs $\{1, \dots, N_J\}$ |
| $\mathcal{E}$ | Realized discrete event stream |

## A.2. Stochastic Distributions and Dynamics

Parameters governing the generation of the event stream $\mathcal{E}$, corresponding to $\theta_{\text{distri}}$ and $\theta_{\text{dyn}}$. Table a-2 details the stochastic and dynamic generation parameters.

*Table a-2.* Stochastic Generation and Dynamic Scenarios

| Symbol | Description |
|---|---|
| *Arrivals and Processing* | |
| $\lambda_{\text{base}}$ | Base arrival rate |
| $\lambda(t)$ | Time-varying arrival intensity function |
| $A, T$ | Modulation amplitude and period for nonstationary arrivals |
| $\Delta t_j$ | Inter-arrival time for job $j$ |
| $k_a, \theta_a$ | Shape and scale parameters for arrival Gamma distribution |
| $p_{j,k}$ | Realized processing time for job $j$, operation $k$ |
| $k_p, \theta_p$ | Shape and scale parameters for processing Gamma distribution |
| $B$ | Batch size sampled from $\mathcal{N}(\mu_B, \sigma_B^2)$ |
| *Dynamic Events ($\theta_{dyn}$)* | |
| $p_{\text{cancel}}$ | Probability of job cancellation |
| $p_{\text{rework}}$ | Probability of rework events |
| $p_{\text{prio}}$ | Probability of priority changes |
| $p_{\text{route}}$ | Probability of dynamic route changes |
| $p_{\text{dd\_chg}}$ | Probability of due-date updates |
| $\Delta_{\text{pm}}$ | Preventive maintenance interval |
| $D_{\text{pm}}$ | Duration of preventive maintenance |

## A.3. Metrics: Targets and Observations

The target metric vector is denoted $\boldsymbol{m}^{\star}$, and the empirically observed counterpart from the simulation is $\boldsymbol{m}^{\text{obs}}$. Table a-3 lists the target and observed metrics.

## A.4. Calibration and Difficulty Modeling

Table a-4 summarizes the calibration and difficulty modeling symbols.

## A.5. Configuration-Layer Specification

**Plant Structure.** The plant component of $\mathcal{I}$ encodes the static plant structure: a finite machine set $\mathcal{M}$, where each machine $m \in \mathcal{M}$ is associated with a speed $v_m$, a partition into machine groups $\mathcal{G}$, and a library of process templates $\mathcal{F}$ describing feasible routes for job families (Dauzère-Pérès et al., 2024). Each template $f \in \mathcal{F}$ specifies an ordered set of operations $\mathcal{K}_f$, together with nominal mean processing times $\mu_{f,o}$ for each operation $o \in \mathcal{K}_f$. An optional job-mix vector $\boldsymbol{w}$ with $\sum_{f \in \mathcal{F}} w_f = 1$ controls how frequently each template is sampled. We summarize the structural part of the configuration as

$$\mathcal{P} = (\mathcal{M}, \mathcal{G}, \mathcal{F}, \boldsymbol{w}). \tag{a-1}$$

**Stochastic Distributions.** Stochastic distributions specify probability models for the stochastic primitives that drive the event stream $\mathcal{E}$. The corresponding collection of distribu-

*Table a-3.* Target and Observed Metrics

| Symbol | Description |
|---|---|
| $\star$ / $^{\mathrm{obs}}$ | Superscripts denoting target / observed values |
| $\rho_{\mathrm{global}}$ | Global system utilization |
| $\rho_g$ | Utilization of machine group $g$ |
| $\chi_{\mathrm{load}}$ | Load imbalance coefficient (CV of group utilizations) |
| $c_a^2$ | Squared coefficient of variation of inter-arrival times |
| $c_p^2$ | Squared coefficient of variation of processing times |
| $\tau$ | Due-date tightness (normalized slack factor) |
| $\delta$ | Disturbance ratio (fraction of capacity lost to outages) |
| $\mathcal{B}$ | Set of bottleneck specifications |
| $b$ | Index for a bottleneck window $b \in \mathcal{B}$ |
| $\rho_b$ | Utilization within bottleneck window $b$ |
| $D_j$ | Final due date for job $j$ |
| $W_j$ | Total realized workload for job $j$ |
| $C_{\mathrm{tot}}$ | Nominal total capacity over horizon $H$ |
| $C_b^{\mathrm{eff}}$ | Effective capacity in bottleneck window $b$ |

*Table a-4.* Difficulty Scoring and Calibration

| Symbol | Description |
|---|---|
| *Schedule Stress Index (SSI)* | |
| $d$ | Scalar difficulty score $(0 - 100)$ |
| $C$ | Congestion difficulty component |
| $P$ | Time-pressure difficulty component |
| $K$ | Structural complexity component |
| $S$ | Disruption intensity component |
| $\rho_{\mathrm{nb}}$ | Natural bottleneck utilization $(\max_g \rho_g^{\mathrm{obs}})$ |
| $\hat{X}$ | Log-normalized value for component $X \in \{C, P, K, S\}$ |
| *Calibration* | |
| $\mathcal{L}$ | Index set of calibrated metrics |
| $e_\ell$ | Normalized error for metric $\ell$ |
| $e$ | Error vector |
| $\mathcal{S}$ | Catalog of adjustment strategies |
| $\mathcal{T}_s$ | Transformation operator for strategy $s$ |
| $a_s$ | Impact sensitivity vector for strategy $s$ |
| $\varepsilon_{\mathrm{bn}}$ | Aggregate bottleneck deviation metric |
| $\ell_2$ | Calibration loss ($L_2$ norm of errors) |

tional specifications is denoted by $\theta_{\mathrm{distri}}$. Inter-arrival intervals $\Delta t_j$ are sampled from a Gamma distribution to control arrival variability (Lassoued et al., 2026), operation processing times $p_{j,k}$ are sampled from a Gamma distribution whose parameters are set to match the corresponding template means (Ghaedy-Heidary et al., 2024), and preventive-maintenance durations $D_{\mathrm{pm}}$ are sampled from a normal distribution. One instantiation is

$$\Delta t_j \sim \Gamma(k_a, \theta_a) \quad p_{j,k} \sim \Gamma(k_p, \theta_p) \quad D_{\mathrm{pm}} \sim \mathcal{N}(\mu_{\mathrm{pm}}, \sigma_{\mathrm{pm}}^2), \quad \text{(a-2)}$$

and thus $\theta_{\mathrm{distri}} = \big((k_a, \theta_a), (k_p, \theta_p), (\mu_{\mathrm{pm}}, \sigma_{\mathrm{pm}})\big)$ in this example. Other stochastic primitives, including batch sizes and repair times, are specified by analogous parametric families.

**Target Metrics.** Target metric levels define the desired performance profile of the generated instances. They include a global utilization target $\rho_{\mathrm{global}}^\star$, squared coefficients of variation for inter-arrival and processing times $c_a^{2\star}$ and $c_p^{2\star}$ (Sherzer et al., 2025), a due-date tightness parameter $\tau^\star$, a load-imbalance coefficient $\chi_{\mathrm{load}}^\star$ (Wang et al., 2024c), a disturbance ratio $\delta^\star$, and optional bottleneck utilization targets $\{\rho_b^\star\}_{b \in \mathcal{B}}$ on designated windows. Here $\mathcal{B}$ denotes a set of bottleneck specifications, where each index $b \in \mathcal{B}$ identifies a time window together with an associated machine group intended to be capacity-constraining. These targets are collected into

$$m^\star = \big(\rho_{\mathrm{global}}^\star, c_a^{2\star}, c_p^{2\star}, \tau^\star, \chi_{\mathrm{load}}^\star, \delta^\star, \{\rho_b^\star\}_{b \in \mathcal{B}}\big), \quad \text{(a-3)}$$

with observed counterparts $m^{\mathrm{obs}}$ computed from the realized event stream $\mathcal{E}$.

**Dynamic Scenarios.** Dynamic-scenario parameters configure stochastic events beyond baseline job arrivals and operation processing. They control event types such as cancellations, rework, priority changes, route changes, due-date updates, and preventive maintenance (An et al., 2023). The dynamic layer is characterized by the following key controls:

$$\theta_{\mathrm{dyn}} = \big(p_{\mathrm{cancel}}, p_{\mathrm{rework}}, p_{\mathrm{prio}}, p_{\mathrm{route}}, p_{\mathrm{dd\_chg}}, \Delta_{\mathrm{pm}}\big), \quad \text{(a-4)}$$

where $p_{\mathrm{cancel}}, p_{\mathrm{rework}}, p_{\mathrm{prio}}, p_{\mathrm{route}}$, and $p_{\mathrm{dd\_chg}}$ control the frequencies of cancellation, rework, priority-change, route-change, and due-date change events, respectively, and $\Delta_{\mathrm{pm}}$ controls the preventive maintenance interval.

## B. Parameter-Space Multi-Objective Calibration Details

This appendix details the design of the parameter-space calibration baseline. Unlike SESC which operates on realized event lists, this approach treats the instance generation configuration as a black-box function $G(x) \to \mathcal{E}$ and optimizes the generative hyperparameters $x$ directly.

### B.1. Problem Formulation

Let $x \in \Omega \subset \mathbb{R}^d$ be a vector of generative hyperparameters. The generation process induces a vector of observed metrics

$m^{\mathrm{obs}}(\boldsymbol{x})$. The calibration task is formulated as a multi-objective optimization problem:

$$\min_{\boldsymbol{x} \in \Omega} \boldsymbol{F}(\boldsymbol{x}) = \big(f_1(\boldsymbol{x}), f_2(\boldsymbol{x}), \ldots, f_K(\boldsymbol{x})\big) \qquad \text{(b-1)}$$

where each objective $f_k$ corresponds to a normalized error for a target metric $m_k^\star$. The objective functions are defined as:

$$f_k(\boldsymbol{x}) = \begin{cases} \dfrac{|m_k^{\mathrm{obs}}(\boldsymbol{x}) - m_k^\star|}{|m_k^\star| + \varepsilon}, & \text{if } |m_k^\star| > 0, \\ |m_k^{\mathrm{obs}}(\boldsymbol{x})|, & \text{if } m_k^\star \approx 0. \end{cases} \qquad \text{(b-2)}$$

Here, $\varepsilon = 10^{-6}$ is a numerical stability constant. The objective vector $\boldsymbol{F}$ typically has dimension $K = 5$ to $7$, covering global utilization, arrival/processing SCV, due-date tightness, disturbance ratio, and optionally load imbalance and bottleneck utilization.

## B.2. Decision Space Specification

We define two variants of the decision space: the Extended 12D Space (used as the full MOO baseline) and the Reduced 5D Space (used in the Hybrid Calibrator).

**Extended 12D Decision Vector.** The extended parameter vector $\boldsymbol{x} \in \mathbb{R}^{12}$ controls all stochastic aspects of the generator. The components and their scaling logic are defined in Table b-1.

*Table b-1.* Decision variables for the Extended MOO Calibrator.

| Index | Parameter | Transformation Logic |
|-------|-----------|----------------------|
| $x_0$ | Arrival Rate Scale | $\lambda_{\mathrm{new}} \leftarrow \lambda_{\mathrm{base}} \cdot x_0$ |
| $x_1$ | Arrival SCV Scale | $c_{a,\mathrm{new}}^2 \leftarrow \min(c_{a,\mathrm{base}}^2 \cdot x_1, 5.0)$ |
| $x_2$ | Process SCV Scale | $c_{p,\mathrm{new}}^2 \leftarrow \min(c_{p,\mathrm{base}}^2 \cdot x_2, 5.0)$ |
| $x_3$ | Due-Date Scale | $\tau_{\mathrm{new}} \leftarrow \mathrm{clip}(\tau_{\mathrm{base}} \cdot x_3, 0.5, 10)$ |
| $x_4$ | Arr. Shape Bias | Fine-tuning gamma shape $k_a$ |
| $x_5$ | Proc. Shape Bias | Fine-tuning gamma shape $k_p$ |
| $x_6$ | Breakdown Scale | Scales disturbance budget $\delta$ |
| $x_7$ | Routing Balance | (Reserved for routing entropy) |
| $x_8$ | Batch Intensity | Scales mean batch size $\mu_B$ |
| $x_9$ | Initial WIP | Scales warm-start job count |
| $x_{10}$ | Job Mix $\alpha$ | $w_f' \propto (w_f)^{x_{10}}$ (controls load balance) |
| $x_{11}$ | Bottleneck Scale | Scales target $\rho_b^\star$ for specific windows |

**Reduced 5D Decision Vector.** For the Hybrid approach, global utilization and disturbance are handled by the constructor. The optimization focuses on the coupled metrics using $\boldsymbol{x} \in \mathbb{R}^5$:

$$\boldsymbol{x}_{\mathrm{hybrid}} = \big(\gamma_{\mathrm{scv\_a}}, \gamma_{\mathrm{scv\_p}}, \gamma_{\mathrm{ddt}}, \gamma_{\rho_{\mathrm{bn}}}, \gamma_{\mathrm{load\_cv}}\big) \qquad \text{(b-3)}$$

where each component acts as a multiplicative scalar on the respective baseline configuration.

## B.3. Optimization and Solution Selection

We employ the NSGA-II to approximate the Pareto front.

- Population: 60 (MOO) or 100 (Hybrid).

- Generations: Fixed 40 generations or adaptive termination based on hypervolume convergence.

- Operators: Simulated Binary Crossover (SBX, $\eta = 15$) and Polynomial Mutation (PM, $\eta = 20$).

**Constraint-Prioritized Selection.** To select a single best configuration $\boldsymbol{x}^\star$ from the non-dominated set $\mathcal{P}$, we apply a lexicographic selection rule based on tolerance satisfaction. Let $\boldsymbol{\theta} \in \mathbb{R}^K$ be the vector of acceptable tolerances. For each solution $\boldsymbol{x}^{(i)} \in \mathcal{P}$, we calculate the number of satisfied constraints:

$$N_{\mathrm{sat}}^{(i)} = \sum_{k=1}^{K} \mathbb{I}\left[ f_k(\boldsymbol{x}^{(i)}) \leq \theta_k \right] \qquad \text{(b-4)}$$

The optimal solution is selected via:

$$i^\star = \arg\max_i \left( N_{\mathrm{sat}}^{(i)}, -\sum_{k=1}^{K} f_k(\boldsymbol{x}^{(i)}) \right) \qquad \text{(b-5)}$$

This ensures that the selected parameter set maximizes the number of metrics within tolerance before minimizing the aggregate error.

# C. Dataset Specifications and Scaling Configurations

This appendix provides detailed specifications for the two primary datasets used in the experimental evaluation: **DynaSched-Grid** and **DynaSched-Sweep**.

## C.1. DynaSched-Grid: Structured Operational Coverage

DynaSched-Grid is designed to provide broad, structured coverage of the dynamic job shop operational space. Instances are generated by taking the Cartesian product of discretized levels for the six primary driver metrics.

To ensure statistical reliability, each configuration in the grid is instantiated with 5 independent random seeds. The base plant structure for these instances consists of 10 machines organized into 4 machine groups, processing 3 distinct job families with an average route length of 4 operations.

## C.2. DynaSched-Sweep: Local Sensitivity Analysis

DynaSched-Sweep focuses on high-resolution local sensitivity analysis. It is constructed by selecting a "centroid"

*Table c-1.* Parameter grid for DynaSched-Grid generation. The full factorial combination yields a diverse set of base configurations.

| Metric | Symbol | Levels |
|---|---|---|
| Global Utilization | $\rho_{\text{global}}^{\star}$ | $\{0.50, 0.65, 0.80, 0.90, 0.95\}$ |
| Due-Date Tightness | $\tau^{\star}$ | $\{2.0, 4.0, 6.0, 8.0, 10.0\}$ |
| Arrival SCV | $c_a^{2\star}$ | $\{0.25, 0.50, 1.0, 2.0, 4.0\}$ |
| Processing SCV | $c_p^{2\star}$ | $\{0.25, 0.50, 1.0, 2.0\}$ |
| Disturbance Ratio | $\delta^{\star}$ | $\{0.0, 0.05, 0.10, 0.15\}$ |
| Load Imbalance | $\chi_{\text{load}}^{\star}$ | $\{0.0, 0.15, 0.30\}$ |

configuration representing a typical dynamic job shop and then varying one or two parameters at a fine granularity while holding others fixed.

**Centroid Configuration:** $\rho_{\text{global}}^{\star} = 0.85$, $\tau^{\star} = 5.0$, $c_a^{2\star} = 1.0$, $c_p^{2\star} = 0.5$, $\delta^{\star} = 0.05$, $\chi_{\text{load}}^{\star} = 0.1$.

**Sweep Dimensions:**

- Utilization Sweep: $\rho_{\text{global}}^{\star} \in [0.4, 0.98]$ with step 0.02.

- Variability Sweep: Jointly varying $c_a^{2\star}, c_p^{2\star} \in [0.1, 5.0]$ on a logarithmic scale.

- Tightness Sweep: $\tau^{\star} \in [1.5, 12.0]$ with step 0.5.

- Disruption Sweep: $\delta^{\star} \in [0.0, 0.25]$ with step 0.01.

## D. Hyperparameter Sensitivity Analysis

This appendix presents a sensitivity analysis of three calibration methods with respect to their key hyperparameters. The goal is to justify the default parameter selections used in Section 6.2 by visualizing trade-offs between calibration accuracy, measured by Relative $L_2$ Error, and computational cost, measured by wall-clock duration.

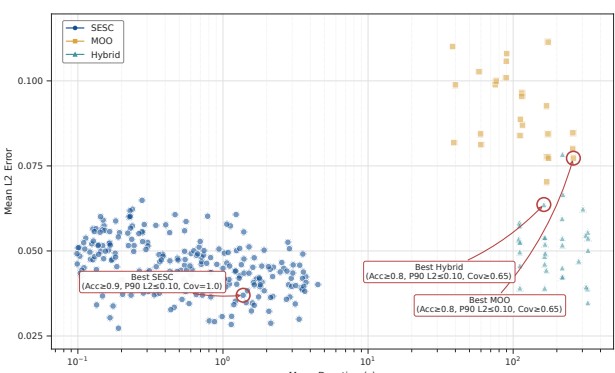

*Figure d-1.* Hyperparameter selection space for the three calibration methods. The highlighted points represent the default configurations selected through feasibility filters based on $Acc$, $P90$, and $Cov$.

Figure d-2 summarizes these relationships. The primary $y$ axis uses bars to show the mode-normalized $L_2$ calibration loss, where lower values indicate better relative calibration. The secondary $y$ axis uses dashed lines to show the mean execution duration in seconds, where lower values indicate faster calibration.

## E. Calibration Success Criteria and Empirical Analysis

This appendix details the mathematical definition of calibration success.

### E.1. Success Criteria Definitions

We define two levels of calibration success Relaxed and Strict to distinguish between general system convergence and precise metric-wise compliance.

**Relaxed Criterion.** A run is considered a relaxed success if the normalized aggregate $L_2$ error falls within the global tolerance $\varepsilon_{\text{tol}}$:

$$\mathbb{I}_{\text{relaxed}} = \mathbb{I}\left[\ell_2(\boldsymbol{m}^{\text{obs}}, \boldsymbol{m}^{\star}) \leq \varepsilon_{\text{tol}}\right] \qquad \text{(e-1)}$$

This criterion indicates that the system's global behavior is aligned with the target profile.

**Strict Criterion.** The Strict criterion requires simultaneous satisfaction of the global bounds and individual relative error bounds for all critical driver metrics $\mathcal{L} = \{\rho_{\text{global}}, c_a^2, c_p^2, \tau, \delta, \chi_{\text{load}}\}$.

$$\mathbb{I}_{\text{strict}} = \mathbb{I}_{\text{relaxed}} \cdot \prod_{l \in \mathcal{L}} \mathbb{I}\left[\text{RE}_l(m_l^{\text{obs}}, m_l^{\star}) \leq \theta_l\right] \qquad \text{(e-2)}$$

The thresholds $\theta_l$ are derived dynamically from $\varepsilon_{\text{tol}}$, ensuring stricter control on fundamental metrics like utilization while allowing reasonable slack for high-variance targets like $c_p^2$.

### E.2. Analysis of Calibration Performance

We analyze the performance of the three calibration modes across three diagnostic perspectives: metric-wise controllability, overall success rates, and error distribution.

#### E.2.1. METRIC CONTROLLABILITY AND FAILURE MODES

Fig. e-1 decomposes the error profile by individual metrics.

- **Processing Variability ($c_p^2$):** This is the most challenging metric, exhibiting the highest relative errors (box heights) and failure rates (lines) across all methods. The MOO method fails to control $c_p^2$ in over 60% of cases, whereas SESC reduces this failure rate to approximately 40%.

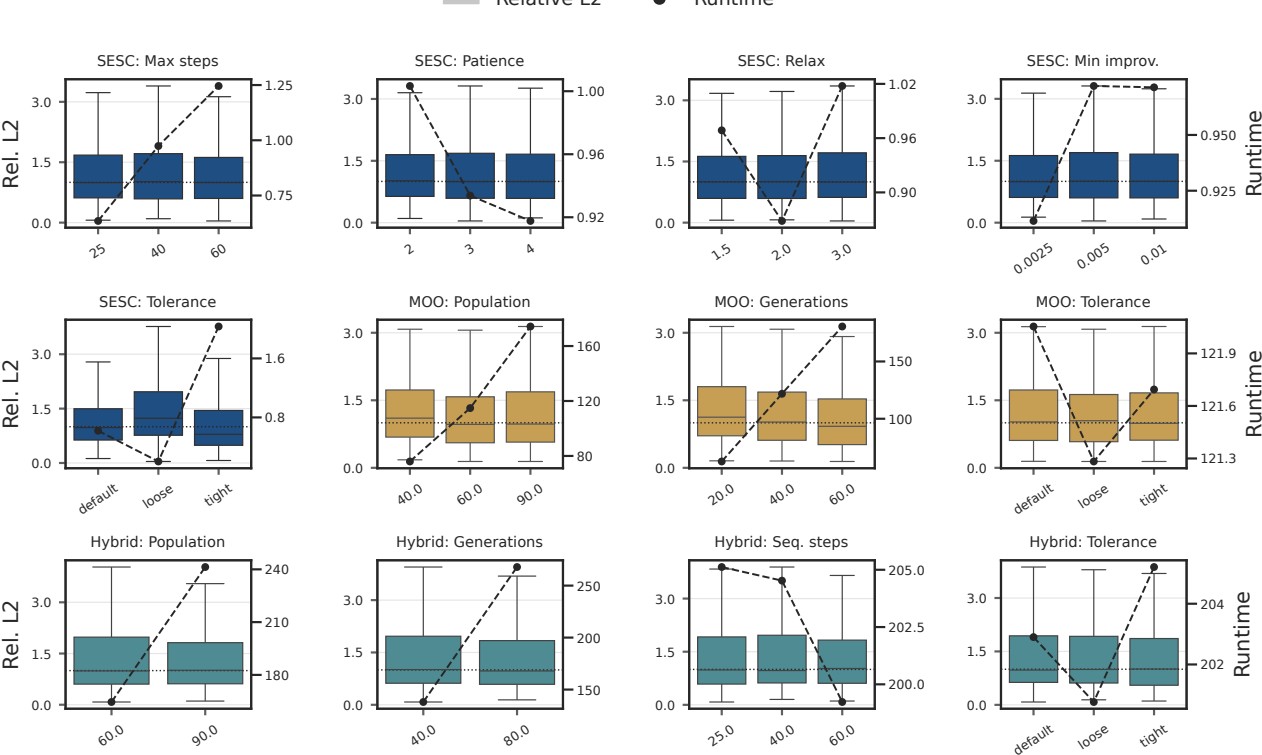

*Figure d-2.* Sensitivity analysis of calibration hyperparameters. Rows group hyperparameter sweeps by calibration mode, as labeled in the figure for SESC, MOO, and Hybrid.

- **Utilization ($\rho_{\mathbf{global}}$) and Tightness ($\tau$):** SESC controls these fundamental metrics with high precision (median error near zero), while optimization-based methods show larger variance.

- **Load Imbalance ($\chi_{\mathbf{load}}$):** The median relative error of Hybrid is low, but the dispersion of SESC is large, which indicates that SESC is difficult to handle under this metric.

#### E.2.2. OVERALL SUCCESS RATES

Fig. e-2 compares the aggregate success rates. SESC achieves the highest relaxed success rate (84%) and a strict success rate comparable to Hybrid (48% vs. 49%), while maintaining a more concentrated low-error distribution. MOO remains clearly weaker under both criteria.

#### E.2.3. ERROR DISTRIBUTION CHARACTERISTICS

The probability density of the final normalized $\ell_2$ calibration error is shown in Fig. e-3. SESC exhibits the strongest concentration in the low-error region, with a sharp density peak around $10^{-2}$–$10^{-1}$, indicating that most runs converge to small calibration errors. MOO displays a broader and more right-shifted distribution, with its density peak occurring at

larger errors and a visibly heavier tail extending toward high-error regimes. Hybrid lies between the two: it improves over MOO by shifting more mass toward lower errors, but its distribution remains less concentrated than SESC. Overall, the density comparison confirms that SESC provides the most stable convergence behavior, while MOO suffers from larger dispersion and Hybrid only partially closes this gap.

## F. Seed Robustness and Stability Analysis

This appendix provides a detailed statistical analysis of calibration stability. To assess robustness, we executed repeated calibration runs ($N = 5$ seeds per configuration) across a diverse subset of scenarios. Fig. e-4 presents a detailed breakdown of calibration performance across eight representative scenarios, ranging from simple static queues to complex dynamic bottlenecks.

**Static and Low-Stress Scenarios.** In simpler cases, all three methods perform relatively well, although SESC maintains a tighter error distribution.

**Dynamic and High-Stress Scenarios.** The performance gap widens significantly in complex scenarios.

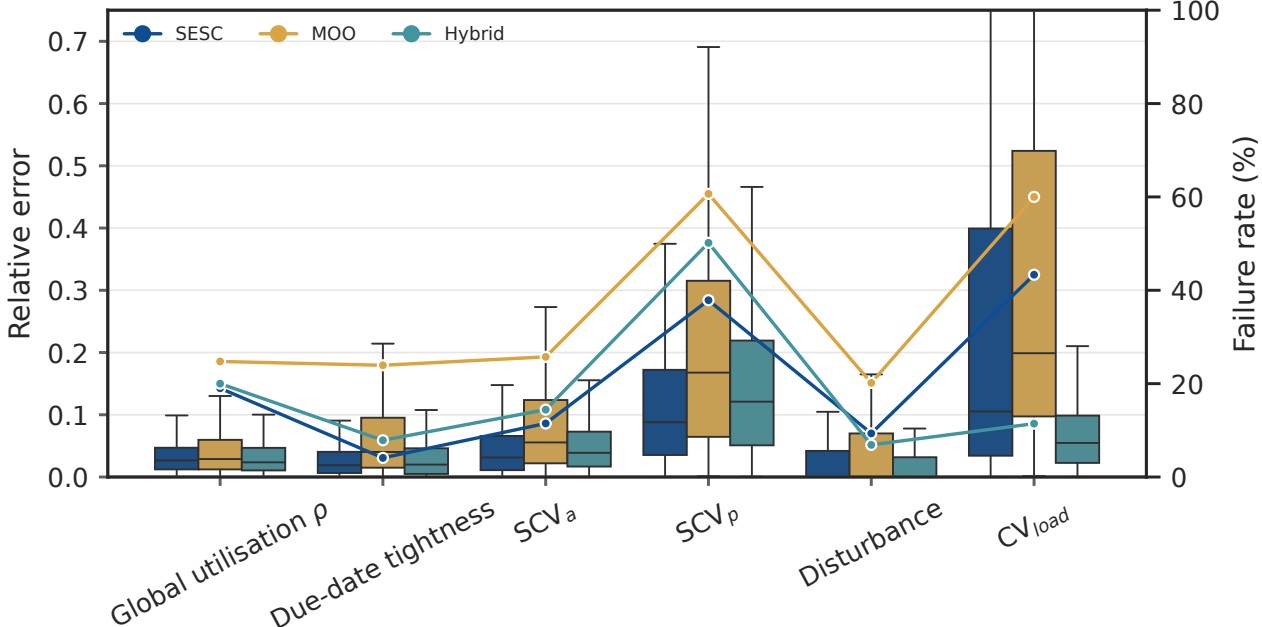

*Figure e-1.* Metric controllability (box plots) and specific failure rates (lines). Processing SCV ($c_p^2$) presents the highest difficulty, while Sequential calibration minimizes failure rates across most dimensions.

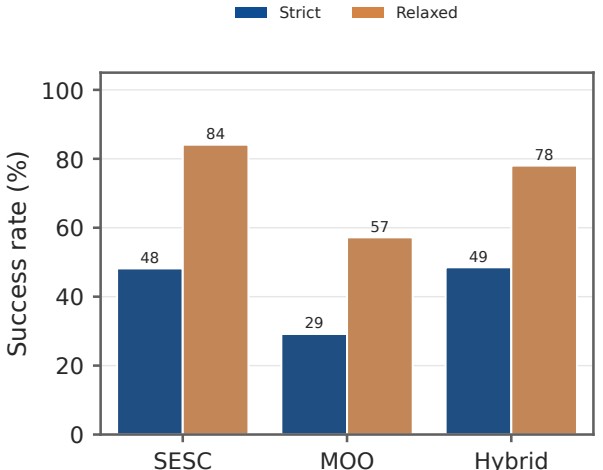

*Figure e-2.* Overall calibration success rates by mode.

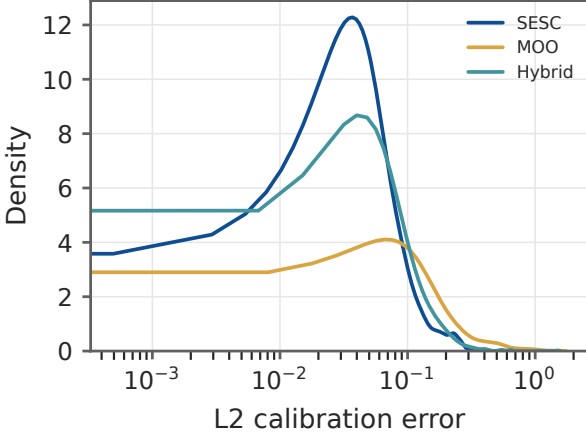

*Figure e-3.* Distribution of final normalized $\ell_2$ calibration error under different calibration methods.

- Dynamic Baselines: MOO calibrator exhibits extreme instability. This suggests that the parameter-space optimizer struggles to find a generalized configuration that accommodates high-intensity dynamic events.

- Bottleneck Constraints: SESC consistently achieves errors below $0.05$. However, MOO shows a collapsed performance boxplot shifted significantly higher, indicating a systematic inability to satisfy the specific bottleneck utilization targets alongside global constraints.

These case studies confirm that while parameter-space optimization can handle basic instances, the direct event-space manipulation used in the SESC is essential for robustly handling the structural constraints and dynamic complexities.

## G. Scaling Experiment Analysis

This appendix provides a detailed analysis of the scaling experiments. We examine how calibration performance varies with problem size, structural complexity, and specific

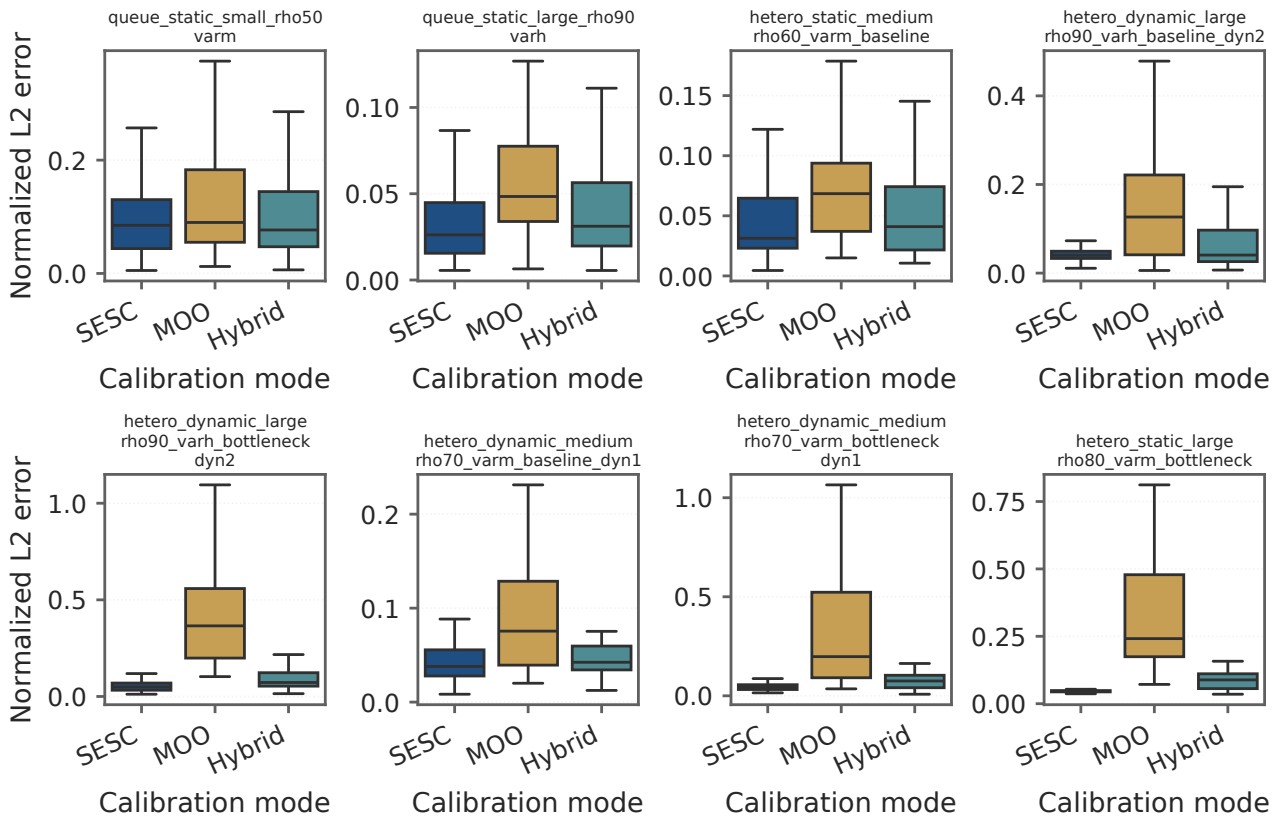

*Figure e-4.* Per-scenario calibration error distributions across 5 random seeds.

metric constraints.

## G.1. Scaling Behavior and Complexity Analysis

We use a static scale-jobs family in which the total job count $N_J$ and horizon $H$ increase proportionally, keeping the nominal arrival rate $\lambda \approx N_J/H$ approximately constant. Only $(N_J, H)$ are rescaled; machine layouts and routing templates remain identical.

*Table g-1.* Level, job count and horizon of scaling configurations.

| Level | $N_J$ | $H$ |
|---|---|---|
| Small | 200 | 1,107.5 |
| Medium | 400 | 2,214.9 |
| Large | 800 | 4,429.9 |
| Extra-Large | 1,600 | 8,859.7 |

The performance of SESC is evaluated across varying problem dimensions and structural complexities. Fig. g-1 shows that increasing problem scale yields lower normalized calibration error but higher wall-clock runtime, indicating improved accuracy at growing computational cost. This analysis assesses the stability of the refinement operators as the density of the event stream increases.

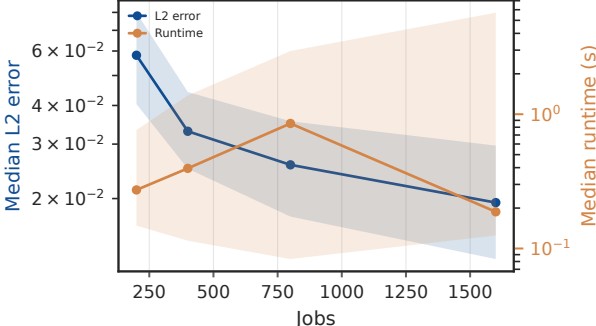

*Figure g-1.* Calibration performance as a function of problem size.

## G.2. Complexity Component Analysis

To study dynamic complexity independently of scale, we fix the scale to $N_J = 200$ and $H \approx 1,107.5$ and vary only the activation of dynamic event types via $\theta_{\mathrm{dyn}}$. All configurations share the same static plant and target performance parameters. Arrivals follow the same periodic pattern, with $\lambda(t)$ periodic with amplitude $A = 0.5$ and period $T = H/3$.

We consider five dynamic configurations. Baseline-Static disables all dynamic scenarios by setting $\theta_{\mathrm{dyn}} = \mathbf{0}$, so can-

cellation, rework, batch arrivals, preventive maintenance, route changes, and due-date changes have zero probability. Batch-Only adds batched arrivals to Baseline-Static with $p_{\text{batch}} = 0.10$ and mean batch size $\mu_B = 3$. PM-Only adds periodic preventive maintenance on all machines with interval $\Delta_{\text{pm}} \approx H/8$ and duration $D_{\text{pm}} \sim \mathcal{N}(10, 2^2)$. Route-Only introduces routing disruptions through route-change events with probability $p_{\text{route}} = 0.01$. Full-Complexity enables a combination of stochastic mechanisms, as summarized in Table g-2. Fig. g-2 isolates the impact of different dynamic complexity features.

*Table g-2.* Dynamic mechanisms enabled in the Full-Complexity configuration.

| Mechanism | Parameter(s) | Setting |
| --- | --- | --- |
| Machine availability / breakdowns | $\delta^\star$ | 0.10 |
| Preventive maintenance | $\Delta_{\text{pm}}, D_{\text{pm}}$ | $\Delta_{\text{pm}} \approx H/8;$ $D_{\text{pm}} \sim \mathcal{N}(10, 2^2)$ |
| Batch arrivals | $p_{\text{batch}}, \mu_B$ | $p_{\text{batch}} = 0.10; \mu_B = 3$ |
| Processing-time changes | $p_{\text{ptime}}$ | $p_{\text{ptime}} = 0.05;$ multipliers $\in \{0.7, 0.9, 1.2, 1.5\}$ |
| Job cancellations | $p_{\text{cancel}}$ | 0.02 |
| Rework | $p_{\text{rework}}$ | 0.01 |
| Route changes | $p_{\text{route}}$ | 0.01 |
| Due-date changes | $p_{\text{due}}$ | $p_{\text{due}} = 0.02;$ tightening ratio 0.5 |
| Priority promotions | $p_{\text{prio}}$ | 0.10 |

- Calibration Error Fig. g-2a: The batch-only scenario attains the highest median calibration error, while the full-complexity scenario exhibits a comparatively large spread, indicating higher variability across instances when multiple dynamic mechanisms are enabled. The pm-only scenario is closer to the baseline than batch-only, suggesting that periodic maintenance is less disruptive to calibration than batched arrivals under these settings.

- Computational Cost Fig. g-2b: Runtime varies substantially across scenario types, with route-only exhibiting the highest median and the widest spread. Its median runtime is clearly higher than the Baseline median, while the remaining scenarios remain closer to Baseline-level costs.

## G.3. Scaling Heatmaps: Size vs. Difficulty

Fig. g-3 presents a dual-view heatmap of calibration performance across the problem size $N_J$ and difficulty scores.

- Median Error Fig. g-3a: Median errors are highest at the smallest scale $N_J = 200$ across difficulty categories, and decrease for larger problem sizes. The reduction is most pronounced for high instances, whose median error drops substantially as $N_J$ increases.

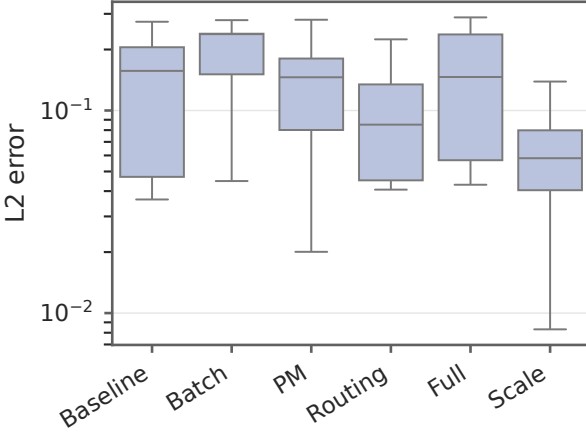

*(a)* Calibration Error ($L_2$, log scale)

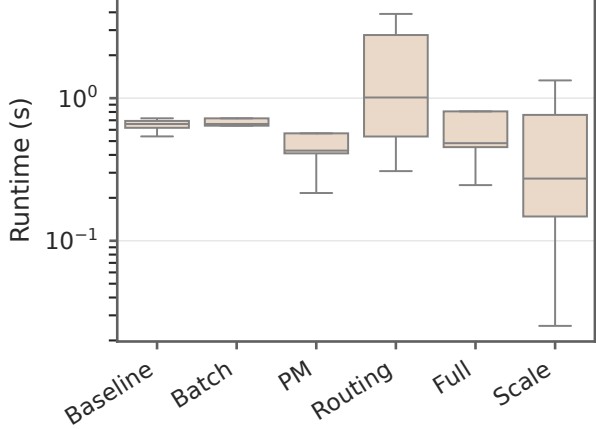

*(b)* Computational Cost (Runtime, log scale)

*Figure g-2.* Impact of complexity components on calibration performance.

- Success Rate Fig. g-3b: Success rates increase with problem size across all difficulty categories. The weakest regime is the smallest scale $N_J = 200$, especially for Hard instances, while success reaches near-perfect levels at $N_J = 1600$.

## G.4. Metric-Wise Scaling Robustness

Fig. g-4 decomposes the scaling behavior by individual driver metrics.

- Robust Metrics Fig. g-4a: Global utilization $\rho_{\text{global}}$ exhibits consistently low relative error across all problem sizes. The due-date related metric ddt also improves with scale and remains low at larger sizes, indicating that these system-level targets are generally easier to control than variability metrics such as $\text{SCV}_p$.

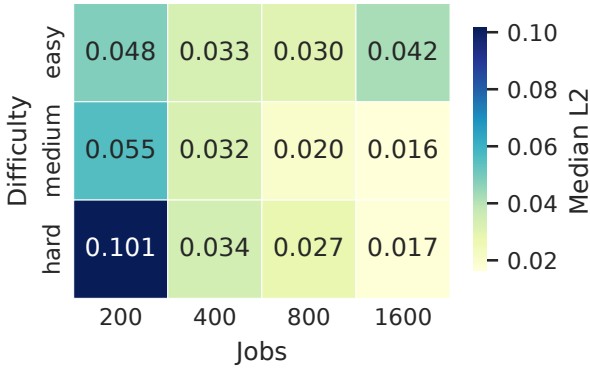

*(a)* Median Calibration Error ($L_2$)

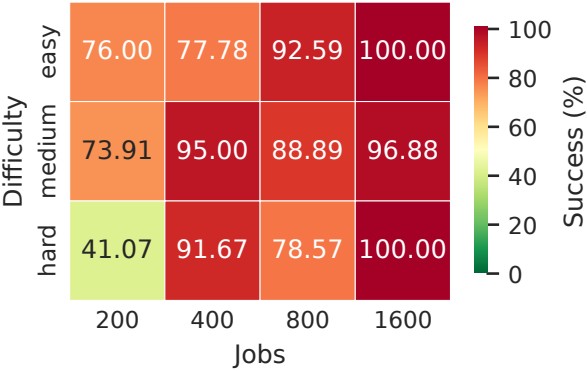

*(b)* Calibration Success Rate (%)

*Figure g-3.* Heatmaps by problem size and difficulty.

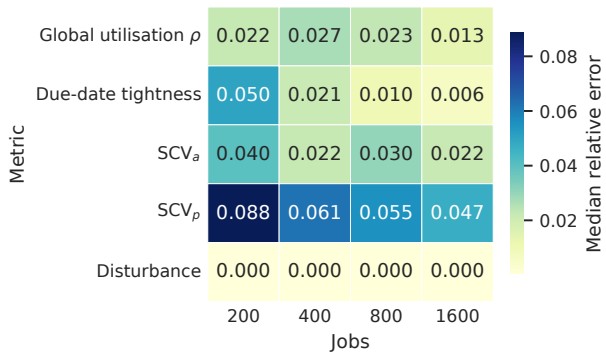

*(a)* Median Relative Error

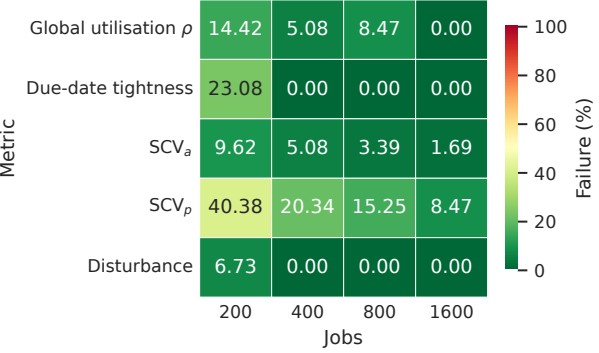

*(b)* Metric Failure Rate (%)

*Figure g-4.* Metric-wise robustness across scales.

## H. Representative Subset Selection for LLM Evaluation

### H.1. Selection Objective and Feature Engineering

Given the scale of the **DynaSched-Grid** and **DynaSched-Sweep** benchmarks, a representative subset **DynaSched-Subset** is constructed to facilitate tractable LLM evaluation while preserving the structural diversity and difficulty variance of the full instance space. The selection process maps each calibrated run $i$ to a $d$-dimensional feature vector $\boldsymbol{z}_i \in \mathbb{R}^d$. The feature set includes the six primary driver metrics: $\rho_{\text{global}}^{\text{obs}}, \tau^{\text{obs}}, c_a^{2,\text{obs}}, c_p^{2,\text{obs}}, \delta^{\text{obs}}, \chi_{\text{load}}^{\text{obs}}$.

To incorporate the difficulty scores into the selection manifold, the difficulty score $d_i$ is transformed into a relative percentile rank $r_i \in (0, 1]$ within its respective dataset. The final feature vector $\boldsymbol{x}_i$ is constructed by concatenating the metrics and the difficulty rank. To ensure an isotropic distance metric for subsequent clustering, each dimension $j$ is standardized via Z-score normalization:

$$z_{ij} = \frac{x_{ij} - \mu_j}{\sigma_j}, \tag{h-1}$$

where $\mu_j$ and $\sigma_j$ denote the mean and standard deviation of

- Sensitive Metrics Fig. g-4b: Processing variability $\text{SCV}_p$ exhibits the highest failure rates across scales, indicating that second-order variability targets are more difficult to control than system-level metrics. Failure rates generally decrease as $N_J$ increases, but $\text{SCV}_p$ remains the most challenging dimension even at larger scales.

### G.5. SESC Operator Ablation

Table g-3 summarizes the aggregate mean and p90 deltas for the main operator removals and minimal operator subsets. Removing slack scaling causes the largest degradation in normalized $L_2$ error. Minimal operator subsets remain competitive on average but lose tail robustness.

Slack scaling is therefore the clearest average-case convergence driver. Operators with small or negative mean deltas still show positive p90 deltas, indicating that they mainly protect high-stress tails. Minimal subsets recover average behavior but consistently degrade tail robustness.

*Table g-3.* SESC ablation deltas relative to the full operator set. Positive values indicate worse normalized $L_2$ calibration error.

| Variant | Mean delta | p90 delta |
| --- | --- | --- |
| No slack scaling | +0.0225 | +0.0902 |
| No bottleneck engineering | +0.0007 | +0.0056 |
| No due relax | -0.0002 | +0.0045 |
| No processing-time resample | -0.0077 | +0.0011 |
| No arrival structure | -0.0124 | +0.0078 |
| Minimal: arrival + due | +0.0036 | +0.0528 |
| Minimal: arrival + due + bottleneck | +0.0075 | +0.0603 |
| Minimal: arrival + due + processing-time | +0.0122 | +0.0540 |

the $j$-th feature across the candidate pool.

## H.2. Greedy $k$-center Diversity Sampling

The subset selection is formulated as a $k$-center problem, which seeks to identify a subset of centers $C \subset \mathcal{X}$ with $|C| = k$ such that the maximum distance from any point in the manifold to its nearest center is minimized:

$$\min_{C \subseteq \mathcal{X}, |C|=k} \max_{x \in \mathcal{X}} \min_{c \in C} \|z_x - z_c\|_2. \qquad \text{(h-2)}$$

To find an approximate solution, a farthest-point greedy heuristic is employed. The algorithm is initialized by selecting the instance with the maximum difficulty score as the first center ($c_1 = \arg\max_i d_i$). In each subsequent iteration $t \in \{2, \dots, k\}$, the next center $c_t$ is chosen as the point that maximizes the distance to the existing set of centers:

$$c_t = \arg\max_{x \in \mathcal{X}} \left( \min_{c \in \{c_1, \dots, c_{t-1}\}} \|z_x - z_c\|_2 \right). \qquad \text{(h-3)}$$

This procedure ensures that the selected subset covers the boundaries of the metric space while maintaining a well-distributed representation of the interior density.

## H.3. Coverage Analysis and Embedding Visualization

We first verify coverage directly in the original driver metrics. Table h-1 compares DynaSched-Subset with the full generated pool through metric ranges and KS statistics. These diagnostics show that DynaSched-Subset preserves the full-set (DynaSchedBench-Grid and DynaSchedBench-Sweep) interquartile range of all driver metrics while remaining small enough for tractable LLM evaluation.

Fig. h-1 provides a two-dimensional visualization of the selected subset overlaying full-set, generated via Principal Component Analysis (PCA) on the standardized metric vectors. The visualization confirms several critical properties of the selection strategy. First, the subset is not concentrated solely in the high-density core of the instance cloud but is distributed across the entire support of the DynaSched-Grid (circles) and DynaSched-Sweep (squares). Second, the use of the farthest-point heuristic is evident as the subset captures extreme points in the embedding space, representing

*Table h-1.* Coverage diagnostics for DynaSched-Subset against the full generated pool.

| Metric | Full range | Subset range | KS |
| --- | --- | --- | --- |
| Difficulty | [4.58, 33.12] | [4.58, 30.47] | 0.175 |
| Utilization | [0.18, 1.43] | [0.31, 0.96] | 0.086 |
| Due-date tight. | [1.11, 10.11] | [1.33, 10.08] | 0.237 |
| Arrival SCV | [0.19, 17.17] | [0.24, 9.02] | 0.251 |
| Processing SCV | [0.11, 6.43] | [0.13, 4.00] | 0.144 |
| Disturbance | [0.00, 0.45] | [0.00, 0.42] | 0.286 |
| Load imbalance | [0.00, 0.65] | [0.00, 0.59] | 0.162 |

boundary conditions such as near-saturation utilization or maximum disturbance intensity. Third, the inclusion of both DynaSched-Grid and DynaSched-Sweep prototypes ensures that the LLM evaluation captures both broad structural transitions and fine-grained sensitivity.

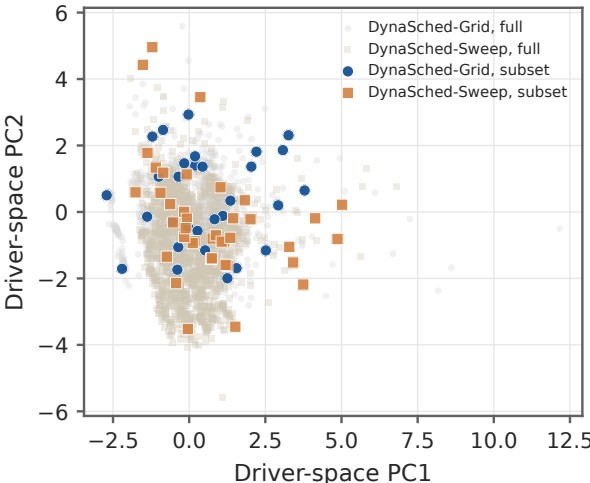

*Figure h-1.* Two-dimensional instance-space embedding overlaying the full-set and the selected LLM subset. The subset selection maintains high coverage across both structured grid points and local sensitivity sweeps.

Fig. h-2 gives a direct distributional check of the scalar difficulty score. The ECDF confirms that the subset retains low, middle, and high difficulty regimes, with a deliberate emphasis on difficult boundary cases.

## I. SSI Robustness and Outcome Validation

This appendix checks whether SSI rankings depend on a particular weighting or normalization choice. We stress-test SSI under component-weight perturbations, internal normalization sweeps, and alternative downstream normalizers. Across the full-set, four component-heavy weightings preserve the baseline ranking with Spearman $\rho \in [0.913, 0.999]$ and Kendall $\tau \in [0.766, 0.984]$.

An 8-step one-at-a-time sweep over SSI normalization caps

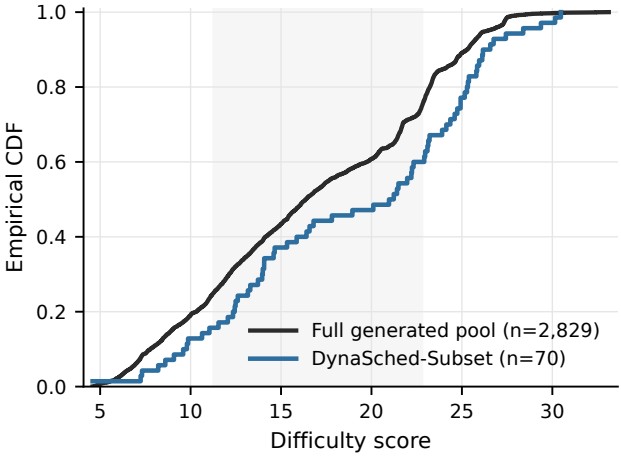

*Figure h-2.* Difficulty-score ECDF of the full-set and DynaSched-Subset.

*Table i-1.* SSI rank stability under component-heavy weight perturbations.

| Scheme | Spearman $\rho$ | Kendall $\tau$ | Mean decile shift |
|---|---|---|---|
| Congestion-heavy | 0.988 | 0.908 | 0.329 |
| Time-pressure-heavy | 0.913 | 0.766 | 0.949 |
| Structure-heavy | 0.999 | 0.984 | 0.086 |
| Disruption-heavy | 0.941 | 0.841 | 0.650 |

and floors also preserves rank stability. The worst-case Spearman $\rho$ remains at least 0.900. Fig. i-1 visualizes the same internal-normalizer sweep at the individual step level. Darker cells indicate stronger agreement with the default SSI ranking.

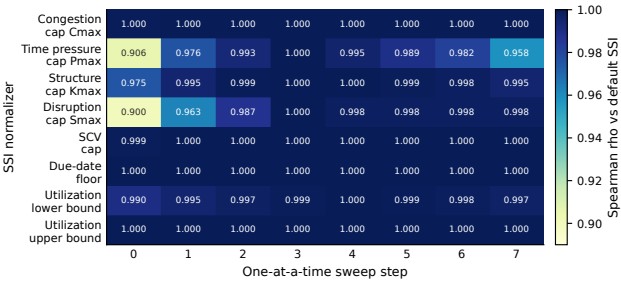

*Figure i-1.* Spearman rank stability of SSI under internal normalizer sweeps.

The same conclusion holds when the downstream difficulty buckets are rebuilt using alternative normalizers. Table i-2 reports the resulting rank agreement and decile shifts.

The alternative normalizers induce moderate decile movement but preserve the broad ordering of easy and difficult instances. We also test whether SSI reflects scheduling difficulty in outcomes.

On the 70-instance evaluation subset, we use $C_{\max}(\text{random}) - C_{\max}(\text{best heuristic})$ as a diffi-

*Table i-2.* Downstream difficulty-bucket stability under alternative SSI normalizers.

| Normalizer | Spearman $\rho$ | Kendall $\tau$ | Mean decile shift |
|---|---|---|---|
| Raw Min-Max | 0.836 | 0.666 | 1.171 |
| Raw Rank-Quantile | 0.852 | 0.666 | 1.143 |
| Raw Z-score-Sigmoid | 0.852 | 0.674 | 1.171 |

culty proxy. SSI correlates significantly with this absolute gap (Spearman $\rho = 0.563$, $p = 3.85 \times 10^{-7}$). The top SSI quintile has a much larger mean gap than the bottom quintile (694.12 vs. 146.71), with an observed mean difference of +547.41 and a 95% bootstrap CI [283.47, 804.00].

Fig. i-2 shows the instance-level relationship between SSI and this outcome gap. The shaded regions mark the bottom and top SSI quintiles used in the tail comparison. This outcome check supports using SSI as a scheduling-difficulty proxy, not only as a calibration diagnostic.

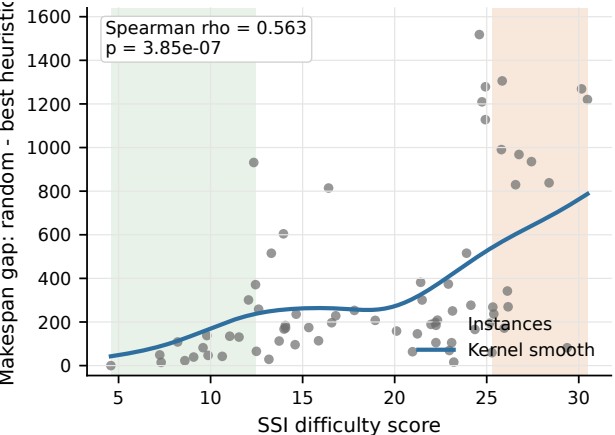

*Figure i-2.* Outcome validation of SSI using the absolute makespan gap between a random policy and the best fixed heuristic.

## J. LLM-based Scheduling Implementation and Diagnostic Analysis

This appendix details the architectural logic of the LLM-based Scheduling and provides a diagnostic interpretation of the empirical results based on the provided performance metrics.

### J.1. Information Observability and Reasoning Logic

The LLM-based Scheduling implementation utilizes a cognitive configuration framework that modulates the density of the state representation and the interaction paradigm between the agent and the environment.

**Observability Levels** ($L_1, L_2, L_3$)**.** The agent's cognition is discretized into three hierarchical levels. Let $O_t$ be the raw

state at time $t$. The encoding function $\mathcal{Z}(O_t, \ell)$ generates a structured observation based on the level $\ell$:

- Level 1: $\mathcal{Z}_1$ includes local machine status, availability timestamps, and nominal operation processing times. This represents the minimal information required for feasible action selection.

- Level 2: $\mathcal{Z}_2$ augments $\mathcal{Z}_1$ with localized statistical counters, including machine group queue lengths, job-level slack factors, and realized event counts (e.g., breakdown frequencies).

- Level 3: $\mathcal{Z}_3$ further incorporates design-layer priors from the input model, such as bottleneck scores and calibration targets. This level tests whether exposing generator-side structural information improves reactive LLM dispatching beyond the statistical summaries in $\mathcal{Z}_2$.

**Refinement and Robustness Mechanisms.** Beyond standard CoT reasoning, two refinement strategies are implemented to evaluate the impact of additional computational budget:

- Reflection: A two-stage process where an initial proposal $a_0$ is subjected to $R$ rounds of self-critique. An independent "auditor" prompt evaluates the action against current queue states, potentially proposing a correction $a'$. The final action is determined over the reflection rounds.

- Best-of-N: A sampling-heavy strategy that draws $N$ independent candidates at a higher temperature. The final decision is reached via majority voting.

**Active Exploration via Tool-Use.** The tool-use mode (L1+Tool) enables an iterative exploration loop. Level 1 agents are permitted to invoke specialized API calls: `inspect_action_details`, which reveals the hidden $L_2/L_3$ information for a specific candidate, and `simulate_action`, which returns an estimated completion time based on a simulated sub-trajectory. The logic implements a multi-turn dialogue where the agent accumulates evidence before final decision.

### J.2. Diagnostic Interpretation of Results

The experimental results, summarized through normalized makespan Mean $C_{max}$, reveal systematic performance behaviors across the four diagnostic perspectives.

**The Observability Paradox.** As illustrated in Fig. j-1-Observability, the relationship between observability and performance is non-monotonic. The L2 CoT configuration

attains the global optimum (0.7%). However, the transition to L3 results in performance degradation (1.7%). This suggests that richer structural priors may create an information-to-action integration burden, especially in tail instances, rather than uniformly improving local dispatching decisions.

**Inefficiency of Active Probing.** Fig. j-1-Probing demonstrates that the L1+Tool configuration (2.0%) fails to improve upon the standard L1 CoT baseline (1.9%). Despite the massive increase in token consumption required for tool-interaction loops, the agent is unable to effectively synthesize the retrieved high-dimensional signals into superior dispatching actions. This identifies a significant "information integration bottleneck" in current agent architectures.

**Reasoning-Performance Diminishing Returns.** The robustness analysis (Fig. j-1-Refinement) indicates that advanced refinement strategies often degrade schedule quality. Both Reflection (1.7%) and Best-of-N (2.7%) perform worse than the standard L2 CoT baseline. This implies that for the specific constraints of dynamic scheduling, deeper reasoning chains do not compensate for the lack of aligned input features and may instead lead to stochastic drift away from optimal local heuristics.

**Mechanism Ablation.** We further run a prompt ablation with Qwen3-8B to identify which part of the richer L3 information contributes to instability. All variants use greedy CoT decoding and are paired by instance. Table j-1 defines the six prompt variants. Table j-2 reports the paired makespan ratios. V3/V1, V4/V1, and V5/V1 each add one mechanism to L2. V6/V2 changes formatting while keeping L3 information fixed. Values above 1 indicate degradation.

*Table j-1.* Prompt variants used in the Observability Paradox mechanism ablation.

| Variant | Observation content | Purpose |
|---|---|---|
| V1 | Standard L2 | Main baseline |
| V2 | Standard L3 | Direct observability contrast |
| V3 | L2 with dummy padding | Tests raw context length |
| V4 | L2 with input model targets | Tests compact target statistics |
| V5 | L2 with bottleneck scores | Tests structural priors |
| V6 | L3 with XML formatting | Tests formatting burden within L3 |

*Table j-2.* Prompt mechanism ablation for the Observability Paradox on 42 complete cases. Ratios are makespan ratios.

| Pair | Mean ratio | Max ratio | Wilcoxon $p$ |
|---|---|---|---|
| V3/V1: length only | 0.998 | 1.029 | 0.427 |
| V4/V1: global targets | 0.999 | 1.018 | 0.278 |
| V5/V1: structural priors | 1.001 | 1.037 | 0.694 |
| V6/V2: L3 formatting | 1.002 | 1.046 | 0.523 |

The single mechanism ablations do not identify a factor with significant mean degradation. L2 plus dummy padding has mean makespan ratio 0.998 relative to L2 and Wilcoxon

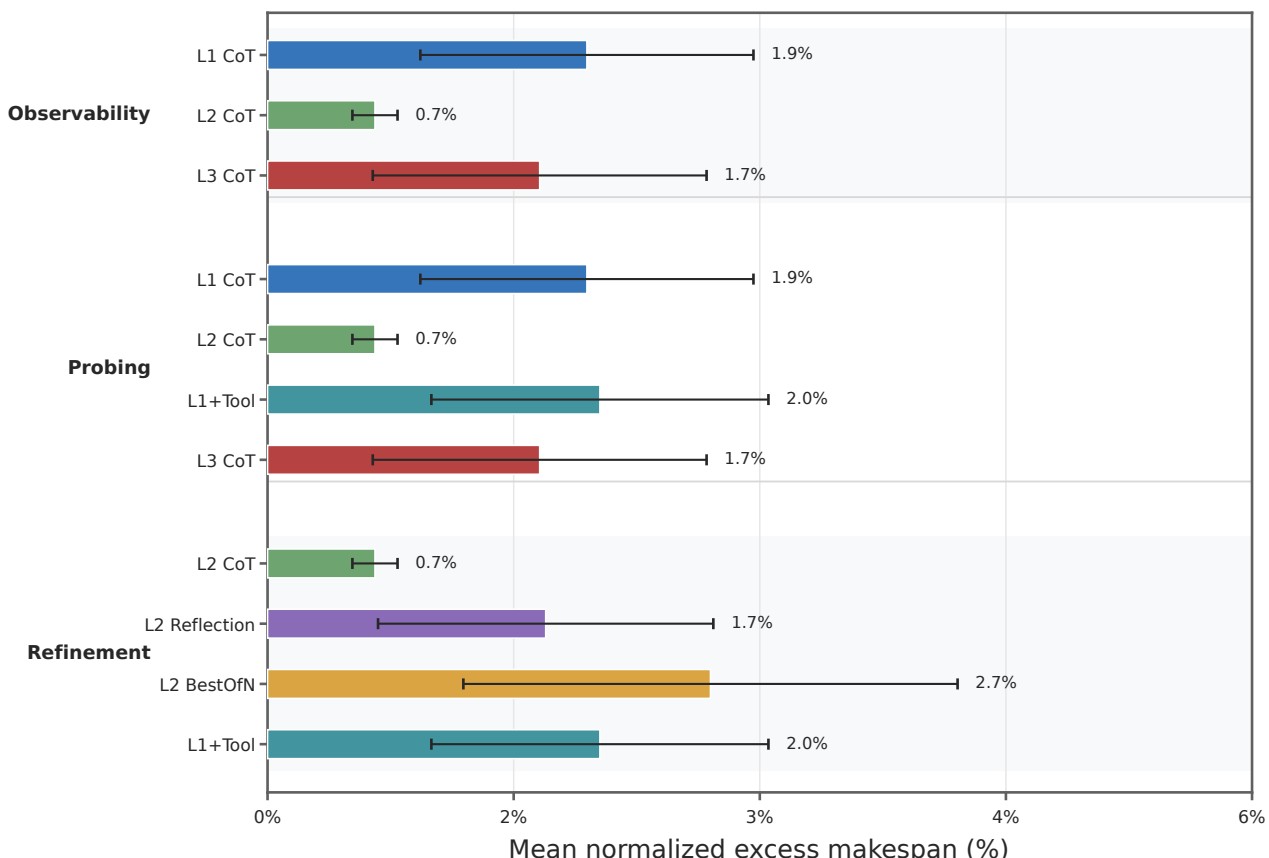

*Figure j-1.* LLM-based Scheduling diagnostic perspectives based on normalized excess makespan.

$p = 0.427$ and compact input model targets are also neutral on average with ratio 0.999 and $p = 0.278$. Structural priors and L3 formatting produce the largest worst case inflations with max ratios 1.037 and 1.046. This supports a tail stability interpretation rather than a simple token length explanation.

Fig. j-2 separates mean, p95, and maximum paired makespan ratios for the mechanism ablations. This layout distinguishes average neutrality from occasional tail inflation. The mechanism evidence therefore narrows the interpretation of the Observability Paradox to tail fragility under richer structured observations.

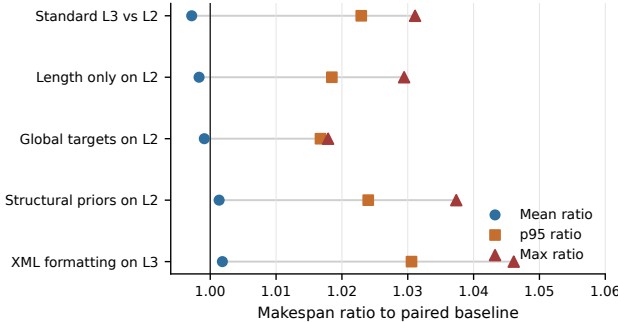

*Figure j-2.* Prompt mechanism ablation for L2 and L3 observability. The main signal is tail fragility.

## K. Detailed Derivations of Generative and Calibration Formulas

This appendix provides the mathematical derivations for the key formulas presented in Section 3 regarding the base arrival rate calculation, stochastic distribution parameterization, and the calibration adjustment operators.

### K.1. Derivation of Base Arrival Rate (Equation 2)

The base arrival rate $\lambda_{\text{base}}$ is calculated to ensure that the system, on average, receives enough work to match the global utilization target $\rho^{\star}_{\text{global}}$.

Let $H$ be the simulation horizon and $\mathcal{M}$ be the set of machines. The total nominal capacity of the shop floor in

machine-time units is:

$$C_{\text{tot}} = H \sum_{m \in \mathcal{M}} v_m, \qquad \text{(k-1)}$$

where $v_m$ is the speed factor of machine $m$.

The target global utilization is defined as the ratio of total workload processed to total capacity:

$$\rho_{\text{global}}^{\star} = \frac{\mathbb{E}[W_{\text{tot}}]}{C_{\text{tot}}}, \qquad \text{(k-2)}$$

where $\mathbb{E}[W_{\text{tot}}]$ is the expected total workload generated during the horizon.

Let $\lambda_{\text{base}}$ be the arrival rate (jobs per unit time). The expected number of jobs arriving in horizon $H$ is $\mathbb{E}[N_J] = \lambda_{\text{base}} H$. The expected work per job, Mean $P$, is the weighted sum of the nominal processing times of the process templates:

$$\text{Mean } P = \sum_{f \in \mathcal{F}} w_f \sum_{o \in \mathcal{K}_f} \mu_{f,o}. \qquad \text{(k-3)}$$

Thus, the total expected workload is:

$$\mathbb{E}[W_{\text{tot}}] = \mathbb{E}[N_J] \cdot \text{Mean } P = (\lambda_{\text{base}} H) \cdot \text{Mean } P. \quad \text{(k-4)}$$

Substituting this into the utilization definition:

$$\rho_{\text{global}}^{\star} = \frac{(\lambda_{\text{base}} H)\text{Mean } P}{H \sum_{m \in \mathcal{M}} v_m} = \frac{\lambda_{\text{base}}\text{Mean } P}{\sum_{m \in \mathcal{M}} v_m}. \qquad \text{(k-5)}$$

Solving for $\lambda_{\text{base}}$ yields Equation 2:

$$\lambda_{\text{base}} = \frac{\rho_{\text{global}}^{\star} \sum_{m \in \mathcal{M}} v_m}{\text{Mean } P}. \qquad \text{(k-6)}$$

## K.2. Parameterization of Gamma Distributions (Equation 5)

The generator uses Gamma distributions $\Gamma(k, \theta)$ for inter-arrival times and processing times to independently control the mean and the SCV denoted as $c^2$.

The probability density function for the Gamma distribution with shape $k$ and scale $\theta$ is:

$$f(x; k, \theta) = \frac{x^{k-1} e^{-x/\theta}}{\theta^k \Gamma(k)}. \qquad \text{(k-7)}$$

The theoretical mean ($\mu$) and variance ($\sigma^2$) are:

$$\mu = k\theta, \qquad \sigma^2 = k\theta^2. \qquad \text{(k-8)}$$

The SCV is defined as:

$$c^2 = \frac{\sigma^2}{\mu^2} = \frac{k\theta^2}{(k\theta)^2} = \frac{k\theta^2}{k^2\theta^2} = \frac{1}{k}. \qquad \text{(k-9)}$$

To match a target SCV $c_a^{2\star}$ for inter-arrival times, we invert the relation:

$$k_a = \frac{1}{c_a^{2\star}}. \qquad \text{(k-10)}$$

To match the target arrival rate $\lambda_{\text{base}}$, the mean inter-arrival time must be $1/\lambda_{\text{base}}$. Therefore:

$$\mu = k_a \theta_a = \frac{1}{\lambda_{\text{base}}} \implies \theta_a = \frac{1}{k_a \lambda_{\text{base}}}. \qquad \text{(k-11)}$$

This confirms the derivation of Equation 5. The same logic applies to processing times $p_{j,k}$, where the mean is $\mu_{f,o}$ and target SCV is $c_p^{2\star}$.

## K.3. Derivation of Bottleneck Capacity Budgets (Equation 12)

We wish to determine how much capacity loss $\Delta C_b$ must be introduced into a specific bottleneck window $b$ to achieve a target utilization $\rho_b^{\star}$.

Let $C_b$ be the nominal gross capacity of machine group $g_b$ in the time window $[s_b, e_b]$:

$$C_b = (e_b - s_b) \sum_{m \in g_b} v_m. \qquad \text{(k-12)}$$

Let $W_b$ be the nominal workload scheduled for this window. The effective capacity $C_b^{\text{eff}}$ after removing downtime is:

$$C_b^{\text{eff}} = C_b - \Delta C_b. \qquad \text{(k-13)}$$

The target definition implies:

$$\rho_b^{\star} = \frac{W_b}{C_b^{\text{eff}}} = \frac{W_b}{C_b - \Delta C_b}. \qquad \text{(k-14)}$$

Rearranging to solve for the required capacity loss $\Delta C_b$:

$$\rho_b^{\star}(C_b - \Delta C_b) = W_b \qquad \text{(k-15)}$$

$$C_b - \Delta C_b = \frac{W_b}{\rho_b^{\star}} \qquad \text{(k-16)}$$

$$\Delta C_b = C_b - \frac{W_b}{\rho_b^{\star}}. \qquad \text{(k-17)}$$

Since we cannot add capacity (negative loss) via breakdown events, we apply the ReLU function $[\cdot]_+ = \max(\cdot, 0)$, yielding Equation 12:

$$\Delta C_b = \left[ C_b - \frac{W_b}{\rho_b^{\star}} \right]_+. \qquad \text{(k-18)}$$

## K.4. Proof of Workload Conservation in Isomorphic Resampling (Equation 35)

The goal of isomorphic resampling is to change the variance of processing times to match $c_p^{2\star}$ without altering the total

realized workload $\sum p_{j,o}$, which would inadvertently drift the global utilization $\rho_{\text{global}}^{\text{obs}}$.

Let $P_{\text{total}} = \sum_{j \in \mathcal{J}} \sum_{o \in \mathcal{K}_{f_j}} p_{j,o}$ be the original total realized workload. We sample new raw processing times $\tilde{p}_{j,o}$ from a Gamma distribution that satisfies the target SCV. The sum of these new samples is $\tilde{P}_{\text{total}} = \sum \tilde{p}_{j,o}$.

We define the scaling factor $\gamma$ as:

$$\gamma = \frac{P_{\text{total}}}{\tilde{P}_{\text{total}}}. \tag{k-19}$$

The final processing times are defined as $p'_{j,o} = \gamma \cdot \tilde{p}_{j,o}$. Calculating the new total workload $P'_{\text{total}}$:

$$P'_{\text{total}} = \sum_{j,o} p'_{j,o} = \sum_{j,o} (\gamma \tilde{p}_{j,o}) = \gamma \sum_{j,o} \tilde{p}_{j,o} \tag{k-20}$$

$$= \left( \frac{P_{\text{total}}}{\tilde{P}_{\text{total}}} \right) \tilde{P}_{\text{total}} = P_{\text{total}}. \tag{k-21}$$

Thus, the total workload is strictly preserved. Note that multiplying a random variable by a scalar constant $\gamma$ does not change its Coefficient of Variation (CV):

$$CV(p') = \frac{\sqrt{\text{Var}(\gamma \tilde{p})}}{\mathbb{E}[\gamma \tilde{p}]} = \frac{\gamma \sqrt{\text{Var}(\tilde{p})}}{\gamma \mathbb{E}[\tilde{p}]} = CV(\tilde{p}). \tag{k-22}$$

Therefore, $p'_{j,o}$ retains the target SCV sampled in $\tilde{p}_{j,o}$ while conserving the global load.

### K.5. Derivation of Bottleneck Duration Adjustment (Equation 36)

This strategy adjusts the duration of downtime events to correct the bottleneck utilization error $\Delta \rho_b$.

Let the aggregate processing speed of the group be $v_{g_b} = \sum_{m \in g_b} v_m$. A downtime event $e$ with duration $\text{dur}(e)$ on a single machine contributes $v_m \cdot \text{dur}(e)$ to the lost capacity. Assuming machines have similar speeds or we approximate using the group aggregate, the total capacity loss is:

$$C_{\text{loss}} \approx v_{g_b} \times (\text{Total Duration of Outages}). \tag{k-23}$$

We need to introduce an additional capacity loss $\Delta C_b$ (derived from the utilization error). The required change in total duration, $\Delta T$, satisfies:

$$\Delta C_b = v_{g_b} \cdot \Delta T \implies \Delta T = \frac{\Delta C_b}{v_{g_b}}. \tag{k-24}$$

The new total duration of events in the window should therefore be the current duration plus the required change:

$$\sum_e \text{dur}(e)' \approx \sum_e \text{dur}(e) + \Delta T = \sum_e \text{dur}(e) + \frac{\Delta C_b}{v_{g_b}}. \tag{k-25}$$

This corresponds to Equation 36.

