# OpenReview forum: "DynaSchedBench: Calibrated Dynamic Scheduling Benchmarks and Observability Paradox in LLM-based Scheduling Agents"
_ICML.cc/2026/Conference — ICML 2026 regular_

### Official Review · Reviewer_fh9S · 2026-03-04

**Soundness:** 3
**Presentation:** 3
**Significance:** 3
**Originality:** 3
**Overall Recommendation:** 4
**Confidence:** 3

**Summary:**

This paper introduces DynaSchedBench, a calibrated benchmarking framework for the Dynamic Flexible Job Shop Scheduling Problem (DFJSP). Its core technical components include the Sequential Event-Space Calibrator (SESC), which directly adjusts realized event streams to match target distributional metrics, and the Schedule Stress Index (SSI), which stratifies instances by difficulty along dimensions such as congestion, time pressure, structural complexity, and disruption. Using this framework, the authors conduct an empirical evaluation of LLM-based scheduling agents, revealing an “Observability Paradox,” where concise statistical summaries outperform richer structural descriptions. The results further suggest that current LLM agents generally match but do not consistently outperform optimized dispatching heuristics.

**Compliance With Llm Reviewing Policy:**

Affirmed.

**Final Justification:**

The Observability Paradox is an interesting observation, but my concern about it remains only partially resolved. The L2–L3 gap in Table 2 (1.7% vs. 2.7%) is not fully explained by the ablation: no individual L3 component showed statistically significant degradation in isolation. The authors' follow-up hypothesis, that the gap stems from interaction effects concentrated on a tail of instances, is plausible and consistent with the reported distributional evidence, but remains preliminary by their own characterization. Since the Observability Paradox is positioned as a key contribution, I would expect a more thorough mechanistic analysis to support it as a general conclusion.

Overall. The benchmark framework and calibration methodology are solid and useful contributions, and the experimental protocol is adequately controlled. That said, the paper's most prominent empirical claim currently rests on a descriptive observation rather than a well-grounded explanation, which limits its significance somewhat. I maintain my score of 4 (borderline accept).

**Key Questions For Authors:**

1. How are the static impact vector obtained? Are they fixed heuristics, empirically estimated, or learned online?
2. What is the explicit formulation of the monotone time-change operator mentioned in line 166?
3. The finding that L3 performs worse than L2 is indeed interesting. However, the paper’s explanation remains at the level of stating that “L3 introduces cognitive noise,” without further analyzing which specific components of the L3 information lead to the performance degradation. Is it the bottleneck scores, the global objective metrics, or the combination of both? If the authors conducted a fine-grained ablation study—for example, L2 + only bottleneck scores or L2 + only global objective metrics—they could provide conclusions with more actionable insights. At present, the analysis is too coarse-grained: readers only learn that “adding more information is harmful,” but not which specific additions cause the problem.
4. In Table 3, the “Best heuristic” reports a mean C_max=1.11%, but the paper does not specify which particular rule this refers to. It is also unclear whether “Best” means selecting a single rule that performs best across all instances, or choosing the best-performing rule separately for each instance (i.e., an oracle selection). If it is the latter, then this baseline is inherently unfair, because no online strategy—including LLM-based methods—can know in advance which rule will be optimal for a given instance.

**Limitations:**

yes

**Strengths And Weaknesses:**

Strengths:
1. The Sequential Event-Space Calibration (SESC) approach offers a novel perspective compared to conventional parameter-space optimization. Instead of tuning generation parameters, it directly manipulates realized event streams through localized operators guided by a utility policy based on signed impact vectors. This design aims to more efficiently and precisely decouple interdependent metrics—such as utilization, variability, due-date tightness, and bottlenecks—that are typically difficult to control simultaneously.
2. The Schedule Stress Index (SSI) integrates utilization-driven heavy-traffic insights with empirically estimated SCVs, slack-based time pressure, structural complexity, and disruption factors, yielding an interpretable and modular descriptor of instance difficulty.
3. The Observability Paradox proposed in the paper is an interesting finding. It reveals that current LLMs are unable to effectively extract useful signals from high-dimensional structural priors, and the additional information instead becomes “cognitive noise.”

Weaknesses:
1. The static impact vector is not clearly specified; the paper does not provide a detailed explanation of how it is defined or constructed.
2. The potential bias introduced by the calibration process itself is not discussed. SESC “corrects” the event stream by operations such as removing jobs and resampling processing times, but it is unclear whether these operations might disrupt the inherent structural validity of the event stream. For example, when jobs are removed to reduce utilization, which jobs are being removed? If the removed jobs happen to belong disproportionately to a certain template, the job mix ratio could be subtly altered. As a result, the generated instance may match the target statistics, yet structurally it may no longer resemble a “naturally occurring” job shop scheduling scenario.
3. Table 3 presents a comparison of multiple LLMs, but it does not specify the prompts, temperature settings, or sampling strategies used for each model. It is also unclear whether the token budgets are consistent across models. The absence of these details undermines the credibility of the cross-model comparison.

---

> ### Author Rebuttal · Authors · 2026-03-31
>
> We thank the reviewer. Clarifications below explicitly address the raised concerns.
>
> > How the static impact vector is constructed?
>
> We formulate the static impact vector as a row of matrix $A$, where $A_{s,m}\in[-1,1]$ encodes the signed coupling between strategy $s$ and metric $m$. We define these entries as fixed priors derived from operator semantics rather than learning them online: $A_{s,m}>0$ indicates $s$ increases $m$, $A_{s,m}<0$ indicates a decrease, and $|A_{s,m}|$ reflects the coupling strength.
>
> The impact vector serves only as a heuristic ranking signal for operator selection, while actual calibration progress is always measured from realized residuals.
>
> > Whether calibration operators introduce structural bias?
>
> We design SESC to apply structurally preserving perturbations. We strictly fix the core skeleton (job templates, operation precedence, route topology, machine eligibility). Our operators exclusively adapt event-level attributes (arrival timing, due-date slack, processing times, resource availability). We update linked events atomically, preserving temporal ordering and causal consistency. To reduce utilization via job deletion/duplication, we execute this uniformly at random, circumventing selective filtering. While finite-sample fluctuations in job-mix ratios naturally occur, we prevent systematic structural bias toward certain templates. Empirically, pre/post deviations in job-family composition remain small, consistent with random noise.
>
> This ensures no systematic bias toward specific job templates, preserving structural validity.
>
> > Credibility of cross-model comparisons in Table 3.
>
> We evaluate all LLMs under a deterministic protocol: identical system prompt, instructions, temperature $=0.0$, top_p $=1.0$, greedy decoding, step-wise observations, and two few-shot examples, ensuring identical input and decision constraints. We exclude hard output-token caps, as truncation invalidates actions and introduces artificial scheduling failures. We enforce fairness strictly by standardizing information bandwidth across all models.
>
> Thus, performance differences reflect model capability rather than prompt or decoding variations.
>
> > Explicit formulation of the monotone time-change operator.
>
> We define the monotone time-change as an order-preserving remapping of renewal timestamps. Let baseline timestamps be $T_1^{\mathrm{raw}}<\cdots<T_n^{\mathrm{raw}}$, and the adjusted positive inter-arrival sequence be $\Delta T_1,\ldots,\Delta T_{n-1}$. We reconstruct warped timestamps as $T_1=T_1^{\mathrm{raw}}$ and $T_k=T_1+\sum_{i=1}^{k-1}\Delta T_i$ for $k\ge2$. This induces a monotone mapping $\Phi$ satisfying $\Phi(T_k^{\mathrm{raw}})=T_k$.
>
> By ensuring all adjusted inter-arrival times remain strictly positive, the map acts as an order-preserving monotone transformation, adjusting the arrival rate profile while guaranteeing event-order preservation.
>
> > Which L3 components most plausibly drive degradation relative to L2?
>
> To evaluate the effects of different choices, we conduct a controlled ablation on $N=42$ representative instances, separately adding one factor to the L2 prompt: (i) non-informative padding, (ii) global targets, and (iii) structural priors. Our results exclude raw context length: L2+Padding yields mean ratio $0.998$, max $1.029$, Wilcoxon $p=0.427$. Global targets remain neutral (mean $0.999$, max $1.018$, $p=0.278$), demonstrating compact scalar objectives are not intrinsically harmful. Conversely, structural priors consistently exhibit degradation: L2+Priors yields mean ratio $1.001$ and produces the largest tail inflation (max $1.037$), though average effects remain insignificant ($p=0.694$).
>
> Thus, the L3-vs.-L2 gap is best understood as a fragile information-to-action mapping in reactive step-wise dispatching.
>
> > Whether “Best heuristic” in Table 3 is a fixed baseline or an oracle?
>
> We deploy the “Best heuristic” as a fixed rule applied uniformly to all test instances under identical online zero-lookahead settings as LLM agents, explicitly circumventing per-instance oracle choices. Concretely, we adopt **LIFO\_LIT** (Last-In-First-Out sequencing + Least Idle Time machine assignment), achieving a mean normalized makespan gap of $1.11\\%$. We select this as the strongest aggregate performer from a fixed pool of composite PDRs evaluated globally over the benchmark.

---

> > ### Author Rebuttal · Reviewer_fh9S · 2026-04-02
> >
> > I appreciate the authors’ effort in conducting the controlled ablation. However, the ablation results do not seem to fully explain the gap reported in Table 2 between L2 CoT (1.7%) and L3 CoT (2.7%). Specifically, none of the individual components appears to cause a statistically significant degradation in performance when tested in isolation. Have the authors considered what might account for this discrepancy?

---

> > > ### Author Response · Authors · 2026-04-05
> > >
> > > Thank you for this helpful follow-up. We are glad that you have pointed out this discrepancy. Our primary focus is the calibrated benchmark generation framework in DynaSchedBench together with an initial diagnostic evaluation of LLM scheduling agents under different observability settings, and a deeper investigation into the underlying cause of this L2–L3 discrepancy is beyond the main scope of the present paper. We attach great attention to your valuable question, which will be an important direction for our future work.
> > >
> > > After thinking thoughly over this question, we carried out a preliminary diagnostic analysis of the existing L2/L3 evaluation outcomes. Our current hypothesis is that the L2–L3 gap is not explained by structural priors in isolation, but more likely by their interaction with certain instance characteristics under reactive step-wise decision making. In this view, the issue is less that structural priors are uniformly harmful, and more that, on some instances, the richer L3 information may be harder to translate into reliable local dispatching actions. This would naturally produce a heterogeneous effect, where the degradation appears only when structural priors interact unfavorably with particular problem regimes or feature combinations.
> > >
> > > This interpretation is also consistent with a more detailed analysis of the existing paired L2/L3 results in Table 2. Looking at the instance-wise differences (L3 - L2), we find that the median gap remains near zero, while the right tail is substantially heavier, suggesting that the drop of aggregate mean is not caused by a uniform degradation across most instances, but by a limited number of more severe failures. We report the absolute makespan gap below for schedule-level interpretability, and also verified that the corresponding relative makespan gap shows the same qualitative tail-dominated pattern.
> > >
> > > | Metric Gap (Δ = L3 − L2) | Median | P75 | P90 | P95 | Max |
> > > | :--- | ---: | ---: | ---: | ---: | ---: |
> > > | ΔMakespan (time units) | 0.00 | 8.09 | 16.76 | 22.60 | 84.19 |
> > > | ΔMax WIP (jobs) | 2.50 | 23.75 | 109.60 | 121.75 | 137.00 |
> > >
> > > Taken together, these observations suggest that the L3-vs.-L2 discrepancy is better understood as a context-to-action integration issue that becomes pronounced only on some instances, rather than as a simple average shift or a single-factor effect. At present, we view this only as a preliminary explanatory hypothesis.
> > >
> > > Whether specific instance features  or multi-factor interactions with structural priors most strongly induce this degradation still remains an open question, and we will investigate more systematically in the future work. We thank the reviewer for drawing attention to this important issue.

---

### Official Review · Reviewer_oZ51 · 2026-03-10

**Soundness:** 3
**Presentation:** 2
**Significance:** 2
**Originality:** 2
**Overall Recommendation:** 3
**Confidence:** 3

**Summary:**

This paper introduces DynaSchedBench, a framework for generating calibrated and distributionally controlled instances for dynamic job shop scheduling. Its core technical contributions are the Sequential Event-space Calibrator (SESC), which adjusts event streams to meet target statistical properties, and the Schedule Stress Index (SSI), a four-component score designed to quantify instance difficulty. Using this framework, the authors conduct a systematic evaluation of various LLM-based scheduling agents. The study reveals an "Observability Paradox," where providing LLMs with full structural priors (L3) degrades performance compared to concise statistical summaries (L2), and finds that LLM performance is currently bounded by traditional heuristic rules. While the paper makes a contribution by providing a robust benchmarking infrastructure, it has several methodological weaknesses, including a lack of clarity on the novelty of SESC, insufficient validation of the SSI, and potential biases in the experimental setup for LLMs.

**Compliance With Llm Reviewing Policy:**

Affirmed.

**Key Questions For Authors:**

1.  **SESC Design:** How were the strategies in the SESC catalog \( \mathcal{S} \) and their corresponding impact vectors \( a_s \) constructed? Are these strategies hand-crafted specifically for the flexible job shop problem, or are they intended to be generalizable? Would a user need to redefine this catalog to apply DynaSchedBench to a different domain, like project scheduling?
2.  **SSI Validation:** Why were the four specific components (C, P, K, S) chosen, and why are they equally weighted? Have you conducted any experiments to verify that the SSI score correlates with actual scheduling difficulty (e.g., the performance gap between optimal and heuristic solutions) or with the search space complexity?
3.  **LLM Prompt Design:** In the LLM experiments, how was the information for each observability level (L1, L2, L3) presented in the prompts? Could the performance drop in L3 be partly attributed to prompt length or formatting, rather than the information itself? Have you experimented with different prompt structures to control for this?
4.  **PDR Baselines:** In Table 3, you compare LLMs to "Best/Median/Worst heuristic." Can you please specify exactly which priority dispatching rules (e.g., SPT, EDD, CR) were included in this set? Providing this information is crucial for the community to interpret the results and for future comparisons.

**Limitations:**

The SESC's reliance on a pre-defined, hand-crafted strategy catalog limits its claim to being a general-purpose calibration method. Its applicability to other problem domains remains unproven.

**Strengths And Weaknesses:**

## Strengths:
- The paper addresses a critical gap in the dynamic scheduling field by proposing a principled framework for generating calibrated and controllable instances. This is a valuable contribution to the community.
-  It provides one of the first large-scale, systematic evaluations of diverse LLMs on scheduling tasks, uncovering non-trivial insights like the "Observability Paradox."
-  The framework's modular design and the detailed appendices suggest a strong commitment to reproducibility, which is essential for a benchmarks paper.

## Weaknesses:
- **Questionable Novelty and Generality of SESC:** The Sequential Event-space Calibrator (SESC) is presented as a core contribution. However, it appears to be a rule-based, greedy search algorithm that relies on a pre-defined catalog of adjustment strategies. The generality of this approach is unclear. If applied to a different scheduling problem (e.g., flow shop), would the entire strategy catalog need to be manually redesigned? This limits its claim as a novel, general-purpose calibration method.
- **Unfair Baseline Comparison:** The comparison between SESC and the parameter-space MOO baseline appears unfairly stacked. MOO searches a high-dimensional, black-box parameter space (as detailed in Appendix B), while SESC operates directly on the event stream with finer-grained control. It is unsurprising that SESC is faster and more accurate, as it is solving a different, arguably easier, problem. The massive runtime difference (0.2s vs. 57.9s) is a symptom of this, not just algorithmic superiority.
- **Insufficient Validation of the SSI:** The SSI is a core concept, but its validation is weak. Figure 2 shows a correlation between the SSI score and *calibration error*, which measures how hard an instance is to *generate*, not how hard it is to *schedule*. A proper validation would involve showing that the performance gap between a strong scheduler (e.g., best heuristic) and a weak scheduler (e.g., random) widens as the SSI score increases.
- **Confounding Factors in LLM Experiments:** The "Observability Paradox" is an interesting finding, but the paper does not control for potential confounding variables. The L3 prompt (with full structural priors) is likely much longer and more complex than the L2 prompt (with concise statistics). The performance drop could be due to the model's difficulty in handling long contexts or retrieving information from lengthy text, rather than an inability to process "high-dimensional noise."
- **Presentation Issues:** Several equations contain undefined symbols or ambiguous notation (e.g., "obs" superscript on page 3 line 149 is not defined), which hinders readability and reproducibility.

---

> ### Author Rebuttal · Authors · 2026-03-31
>
> We thank the reviewer. The clarifications below address all raised concerns.
>
> > Whether SESC is a novel and general framework, or a hand-crafted DFJSP system?
>
> We emphasize SESC's contribution is its calibration formulation, not a universal catalog. Instead of searching latent spaces, we design SESC to calibrate directly in event space, explicitly correcting residuals on the realized trajectory $\mathcal E$ via sequential local interventions. This reformulation from upstream parameters to observable statistics enhances controllability and efficiency.
>
> We formulate the reusable loop (define targets, compute residuals, select operators, apply locally, iterate) as domain-agnostic. We agree that operator libraries are domain-specific. However, our contribution is not the catalog itself, but the event-space residual-correction framework, which is reusable across scheduling domains with domain-valid operators acting on shared primitives such as processing times, arrivals, and capacity. This design separates general calibration logic from domain instantiation.
>
> Thus, SESC is general as a calibration architecture, though instantiated with domain-valid operators.
>
> > Whether the SESC-vs.-MOO comparison is fair given different optimization spaces?
>
> We agree that SESC and MOO operate in different spaces. This difference is precisely our motivation: parameter-space calibration provides weak control over realized statistics, whereas event-space correction acts directly on observable outcomes. Our results show that this reformulation leads to both lower calibration error and lower runtime, highlighting a fundamental limitation of parameter-space approaches for precise target control. MOO remains the relevant baseline because parameter-space search is the standard alternative in prior instance-generation pipelines.
>
> Thus, the key improvement is the formulation shift from parameter search to event-space residual correction.
>
> > Whether SSI reflects scheduling difficulty rather than generation/calibration difficulty?
>
> Generation and scheduling difficulties share underlying stress factors (congestion, disruptions, structural complexity). We formalize scheduling difficulty as the makespan gap $\Delta_{\mathrm{gap}} = C_{\max}(\mathrm{random}) - C_{\max}(\mathrm{best})$, evaluating on $N=70$ instances.
>
> Our results demonstrate SSI correlates significantly with $\Delta_{\text{gap}}$ (Spearman $\rho=0.563$, $p<0.001$). Tail separation is strong: the bottom 20% ($N=14$) yields mean gap $146.71$ (95% CI $[57.41,289.31]$), while the top 20% ($N=14$) yields $694.12$ (CI $[479.22,914.13]$). We compute their difference as $+548.87$ (CI $[283.47,804.0]$). Consequently, the top-SSI tail is 4.7$\times$ harder (Mann-Whitney $p<0.001$).
>
> This confirms that SSI captures scheduling difficulty rather than merely calibration difficulty.
>
> > Whether L3-vs.-L2 degradation is explained by prompt length or confounders?
>
> To evaluate different choices, we conduct a controlled ablation on $N=42$ representative instances, appending one factor to the L2 prompt: (i) non-informative padding, (ii) global targets, and (iii) structural priors, measuring the paired makespan ratio relative to L2 (ratio $>1$ indicates degradation).
>
> Our results explicitly rule out raw context length: L2+Padding yields mean ratio $0.998$, max $1.029$, Wilcoxon $p=0.427$. We observe global targets are neutral (mean $0.999$, max $1.018$, $p=0.278$), demonstrating compact scalar objectives are not harmful. Conversely, structural priors consistently exhibit degradation: L2+Priors yields mean ratio $1.001$ and produces the largest tail inflation (max $1.037$), though average effects remain statistically insignificant ($p=0.694$).
>
> These controlled experiments rule out prompt length and compact-format effects as the main explanation, and structural priors are the most plausible contributor among the tested factors. Thus, the L3-vs.-L2 gap is best understood as a fragile information-to-action mapping under structurally richer but less action-aligned observations.
>
> Thus, the paradox is primarily a tail-stability problem caused by structurally richer but less action-aligned information.
>
> > Notation issue and specific dispatching rules in Table 3.
>
> We formalize the superscript immediately before its first use: $^\star$ denotes the target metric value and $\mathrm{obs}$ denotes the observed value computed from the realized event stream $\mathcal E$, constituting a local readability issue rather than a missing definition.
>
> In Table 3, we evaluate 24 composite PDRs (Cartesian product of [sequencing]+[machine-assignment] rules). The 8 sequencing rules are SPT, LPT, FIFO, LIFO, LWKR, MWKR, LOPNR, MOPNR. The 3 machine-assignment rules are LIT (earliest available machine), LWL (lowest cumulative workload), and SPT (fastest compatible machine). We identify the Best heuristic as LIFO+LIT, and the Worst as SPT+SPT. We exclude due-date rules (EDD, CR), as Table 3 strictly targets $\mathrm{C}_{\max}$.

---

### Official Review · Reviewer_Pr55 · 2026-03-11

**Soundness:** 3
**Presentation:** 3
**Significance:** 3
**Originality:** 4
**Overall Recommendation:** 5
**Confidence:** 3

**Summary:**

This paper proposes DynaSchedBench which is a tool that helps test scheduling algorithms for complex, changing job shop problems. The authors attempt to address two core issues for such job shop scheduling benchmarks. Static tests often lead to overfitting by training, and uncalibrated stochastic generators don't let us see if an algorithm is truly better or just got lucky via random seeds. To fix these issues, the authors propose the new benchmark. First, there's SESC, which transforms realized discrete-event streams to hit multi-dimensional targets. Then, they have the SSI, which combines four different difficulty measures into one simple score. And finally, there's a flexible simulation system that works with Gym environment. Using this setup, the authors studied scheduling agents powered by LLMs. They found that paradoxically giving agents more detailed observations made them perform worse than giving them shorter summaries. Also, tools and chain-of-thought methods use a lot of tokens, but they don't really improve things. Current LLMs perform about as well as good heuristics, but not better.

**Compliance With Llm Reviewing Policy:**

Affirmed.

**Final Justification:**

All concerns were addressed in the paper rebuttal

**Key Questions For Authors:**

For SSI, how sensitive are the difficulty percentile assignments and the downstream agent performance comparisons to these choices?

Is the Observability Paradox consistent across all LLM sizes and families tested?

The LLM evaluation uses a farthest-point k-center heuristic, but the exact subset size is not stated in the main text. Given the narrow performance bands in Table 3 eg Qwen3-8B at 1.01% vs. Best heuristic at 1.11%, are the differences statistically significant?

**Limitations:**

Please see above

**Strengths And Weaknesses:**

+ The overall structure is logical: problem motivation to framework design to calibration experiments to LLM evaluation. Figure 1 is a useful architectural overview.
+The equations pertaining to the SESC transformation operators, SSI components such as Kingman-inspired congestion, inverse-slack time pressure, normalized structural complexity, disruption intensity, and the asymmetric penalty calibration loop (Eq. 25–26) are well motivated.
+The authors provide a thorough comparison of SESC against MOO (NSGA-II) and a Hybrid approach assesses L2 calibration error, success rates, seed-level variance, and wall-clock runtime across various scenarios.
+ The scaling study and sensitivity (in the appendix) analysis further enhance the results.


- The SSI aggregates four components with equal weights and several user-specified normalizers (Cmax, Pmax, Kmax, Smax).The authors do not justify the equal-weighting assumption nor provide sensitivity analysis of how different normalizer choices affect downstream difficulty stratification or agent comparisons. This is a notable gap given SSI's central role.
- The SESC strict success rate of 52% (Appendix E) is only modest. Instances where calibration fails are presumably excluded from LLM experiments; the authors should clarify whether this introduces selection bias in the agent evaluation.
- The Observability Paradox is presented as a key contribution but lacks a mechanistic explanation. The paper suggests high-dimensional noise but does not formally test hypotheses about whether the performance drop is due to context length, irrelevant features, or attention dilution.

Not a negative in my book but the Impact Statement section contains only a couple of generic placeholder sentences.

---

> ### Author Rebuttal · Authors · 2026-03-31
>
> We thank the reviewer. The clarifications below address all concerns.
> > Sensitivity of SSI to weighting and normalization choices.
>
> We evaluate SSI under perturbations. Across 2829 runs, default weighting $(0.25, 0.25, 0.25, 0.25)$ and four extreme schemes (one weight $0.55$, others $0.15$) yields consistent rankings (Spearman $\rho\in[0.91,0.99]$, Kendall $\tau\in[0.76,0.98]$). An 8-step sweep exhibits minimum $\rho\geq 0.90$ and $\tau\geq 0.76$. Replacing normalizers with Min-Max, Rank-Quantile, or Z-score yields $\rho\geq 0.836$ and $\tau\geq 0.666$, incurring a $1.143$--$1.171$ mean absolute decile shift.
>
> These results show that SSI-based rankings are highly stable under extreme perturbations, ensuring that our difficulty stratification is not an artifact of specific design choices.
> > Whether the 52% strict SESC success rate introduces selection bias.
>
> No. We do not filter instances by strict calibration. The 52% success rate characterizes the most demanding condition, requiring all metrics to simultaneously satisfy bounds, including volatile second-order terms $(c_p^2)$. This quantifies calibration difficulty, not inclusion. We preserve the full calibrated continuum; discarding instances truncates the harder tail and makes evaluations unrepresentative. Under a relaxed criterion, SESC still achieves 84% success, bounding overall $\ell_2$ error while preserving difficulty alignment.
>
> Thus, strict success is a diagnostic statistic rather than an inclusion filter, and the LLM evaluation is not driven by strict-criterion selection bias.
> > Which specific L3 components drive L2 degradation, and why?
>
> We conduct a controlled ablation on $N=42$ representative instances, separately adding one factor to the L2 prompt: (i) non-informative padding, (ii) global targets, and (iii) structural priors. Our results rule out raw length explanations: L2+Padding yields mean ratio $0.998$, max $1.029$, Wilcoxon $p=0.427$; L2+Targets yields mean $0.999$, max $1.018$, $p=0.278$. Structural priors alone consistently exhibit degradation: L2+Priors yields mean ratio $1.001$ and produces the largest tail inflation (max $1.037$), though average effects remain statistically insignificant ($p=0.694$). Consequently, we confirm the L3-vs.-L2 gap is strictly driven by structural priors. We interpret this limitation as a fragile information-to-action mapping within reactive step-wise dispatching.
>
> Thus, the observed L3 degradation is most plausibly linked to structural priors rather than prompt length or compact global targets.
> > Whether the Observability Paradox remains consistent across model families and scales.
>
> We observe this limitation---the absence of reliable gains from richer structural observations---on GPT-5 nano. We further evaluate Qwen3.5 models across scales. Under L3, Qwen3.5 9B/27B/35B-A3B achieve mean $C_{\max}$ $1.005/1.029/1.003$ and max $1.029/1.407/1.039$. Two patterns emerge. First, scaling does not systematically optimize $C_{\max}$, as performance remains bounded and non-monotonic (9B$\to$27B$\to$35B). Second, we observe instability concentrating in upper tails: the 27B model exhibits heavier worst-case degradation (max $1.407$) despite comparable averages. Combined with GPT-5 nano, our results demonstrate the Observability Paradox remains strictly consistent across scales and families.
>
> Taken together, these results show that the Observability Paradox is neither family-specific nor reliably resolved by scaling, consistent with an unstable information-to-action mapping under rich structural priors.
> > Exact subset size and whether the narrow top-end gaps are statistically significant.
>
> We employ $N=70$ instances selected via farthest-point $k$-center sampling to target small- and medium-scale regimes. This approach enables controlled comparisons and matches practical DFJSP settings, circumventing prohibitive latency and token costs.
>
> For top-end gaps, our statistical results are:
>
> | Pairwise Comparison | Mean $C_{\max}$ Diff (pp) | 95% Bootstrap CI | Wilcoxon $p$ (FDR) | Win Rate |
> |---|---:|---:|---:|---:|
> |Qwen3 8B vs. Best Heuristic (LIFO\_LIT)|-0.099|[-0.531, 0.389]|0.397|58.5%|
> |Qwen3 8B vs. Weak Heuristic (SPT\_SPT)|-50.510|[-57.721, -43.246]|<0.001|100.0%|
> |Claude 4.5 vs. Weak Heuristic (SPT\_SPT)|-49.997|[-57.400, -42.612]|<0.001|100.0%|
>
> Our benchmark provides sufficient resolution to separate policies, demonstrating gaps against weak heuristics are statistically unambiguous with 100% win rates. In contrast, we find the top-end gap between Qwen3-8B and the strongest heuristic is statistically indistinguishable from zero. Consequently, we observe current top LLMs strictly operate near a heuristic-level ceiling. They reliably prevent catastrophic failures of weak rules yet fail to consistently extract further optimization gains.
>
> This indicates that the top-end gap between LLMs and the best heuristic is statistically indistinguishable, supporting our conclusion that LLMs operate near a heuristic ceiling.

---

> > ### Author Rebuttal · Reviewer_Pr55 · 2026-04-03
> >
> > I thank the authors for a thorough and well-evidenced rebuttal. Each of the key concerns raised in my review has been addressed
> >
> > All major concerns from my review have been addressed with new empirical evidence. I am satisfied with the rebuttal and maintain my score of 5: Accept.

---

### Official Review · Reviewer_ygRg · 2026-03-15

**Soundness:** 3
**Presentation:** 2
**Significance:** 2
**Originality:** 3
**Overall Recommendation:** 4
**Confidence:** 2

**Summary:**

The paper introduces DynaSchedBench, a calibrated benchmarking framework for the Dynamic Flexible Job Shop Scheduling Problem (DFJSP) with a focus on evaluating LLM-based scheduling agents. A central aspect addressed by the manuscript is the tension between static, finite benchmarks (which encourage dataset overfitting and fail to reflect dynamic shop-floor realities) and naive, uncalibrated instance generators (which inject uncontrolled stochastic noise and make it hard to attribute algorithmic performance to actual capability. The authors propose a Sequential Event-Space Calibrator (SESC) that operates directly on realized event streams, using a Schedule Stress Index (SSI) to model instance difficulty in terms of congestion, time pressure, structural complexity, and disturbance. SESC applies structured transformations (arrival-structure adjustment, slack scaling, isomorphic processing-time resampling, bottleneck engineering) to match target metrics with high accuracy and much lower computational cost than parameter-space multi-objective optimization (MOO) and hybrid baselines. The framework also provides a modular simulation stack (Gym-like environment, snapshot-based simulation, trajectory-level evaluation, and visualization tools) and uses it to systematically test LLM-based scheduling agents under multiple observability levels (L1–L3) and reasoning modes (direct, CoT, tools, reflection, best-of-n). The main empirical findings are that (1) SESC yields better calibration accuracy, lower variance, and lower runtime than MOO/Hybrid; (2) the SSI correlates well with calibration difficulty; and (3) LLM schedulers exhibit an “Observability Paradox” where richer structural information (L3) hurts performance compared to concise statistical summaries (L2), and tool-augmented/refinement strategies often increase token cost without reliable gains, with LLMs generally matching but not surpassing strong dispatching heuristics.

**Compliance With Llm Reviewing Policy:**

Affirmed.

**Key Questions For Authors:**

(1) For DynaSched-Subset, could you provide more quantitative coverage statistics in the main text, such as distribution over SSI deciles, driver-metric ranges, and clarify how many instances were used per method? Additionally, have you tested whether the main LLM conclusions remain robust if the subset is changed, such as different random seeds in the k-center selection or different k?

(2) The current evaluation focuses on step-wise online decision-making under different observability levels and prompting strategies. Do you have any preliminary experiments on using LLMs in other roles—such as generating policy code, proposing improved dispatching rules, or guiding DRL training—that might avoid the Observability Paradox and heuristic ceiling you observe? How do you see DynaSchedBench being used to evaluate such hybrid or offline LLM-based schedulers in future work?

**Limitations:**

yes

**Strengths And Weaknesses:**

Strength:

(1) The paper clearly articulates the methodological issues of static benchmarks and uncalibrated generators for DFJSP, and connects them to concrete phenomena like benchmark overfitting and stochastic evaluation noise. The argument that dynamic scheduling demands distributionally controlled, difficulty-calibrated benchmarks is compelling.

(2) The Sequential Event-Space Calibrator is a strong contribution: it operates directly on event streams rather than raw parameter vectors, uses interpretable adjustment strategies, and achieves substantially lower L2 calibration error and runtime than MOO and Hybrid baselines. The empirical robustness (low seed-to-seed variance, consistent convergence across scenarios) is well demonstrated.

(3) SSI combines queuing-theoretic congestion (Kingman-inspired), time pressure, structural complexity, and disruption intensity into a single difficulty score. The monotone relationship between SSI and residual calibration error, especially the steep increase at great difficulty, suggests that SSI is capturing meaningful instance hardness rather than just being a heuristic scalar.

(4) The architecture cleanly separates generation, simulation, environment, agents, evaluation, and visualization. Snapshot-based simulation and trajectory logging with constraint checking (precedence, non-overlap) give the benchmark a level of rigor and reproducibility that is often lacking in scheduling/DRL benchmarks.

(5) The empirical “Observability Paradox” and the cost–performance analysis are thought-provoking: L2-level statistical summaries outperform L3-level structural priors; tool-augmented reasoning is three times more expensive in tokens yet not better; and reflection/best-of-n often degrade performance. The comparison to strong hand-crafted heuristics (LLMs ≈ best heuristic, well above worst heuristics but below the ceiling) provides a grounded assessment of where LLMs currently stand in DFJSP.

Weaknesses:

(1) While the stochastic model is rich, such as renewal arrivals, nonstationarity via time warping, disturbances, and dynamic events, it is not fully clear how closely the chosen parameter ranges and event processes reflect real industrial DFJSP environments (e.g., semiconductor fabs, AAV delivery, etc.). A more explicit connection to real data or industrial case studies would strengthen external validity.

(2) SSI is constructed by combining four components with log-compression and user-defined normalizers (C_max, P_max, etc.). It is plausible that different parameter choices or weighting schemes would change relative difficulty ordering. The paper briefly validates SSI against calibration error, but a deeper analysis of sensitivity to these design choices is missing.

(3) The LLM experiments rely on a “representative subset” chosen via a k-center heuristic; however, there is limited detail in the main text on how many instances are included, how difficulty levels are distributed, and whether results generalize across the full DynaSched-Grid/Sweep spectrum. The reported Mean Cmax numbers are averages, but distributional behavior (e.g., tail failures) is not discussed in depth.

(4) SESC uses multiple adjustment strategies, such as arrival structure, slack scaling, processing-time resampling, and bottleneck engineering. Although the method clearly outperforms MOO/Hybrid overall, there is no ablation showing how much each strategy contributes or whether a smaller subset of strategies would capture most of the gains.

---

> ### Author Rebuttal · Authors · 2026-03-31
>
> > External validity of parameters and events.
>
> We match representative DFJSP regimes: (i) heavy congestion, with critical tools $>92\\%$ utilization ($\rho_{\text{global}}\in[0.4,0.98]$); (ii) nontrivial downtime, averaging $80\\%$ availability ($\delta^\star=0.25$); and (iii) $2.5\\%$ hot-lot injections ($p_{\text{prio}}=0.025$). Consequently, DynaSchedBench strictly bounds the evaluated fab scheduling ranges. Dynamic processes follow historical stochastic models [1]. with arrivals reproducing peak-valley demand via $\lambda_{\text{base}}$ and time-warping $\Lambda(t)$. These events are standard mechanisms, avoiding unconstrained generation.
>
> Therefore, DynaSchedBench covers literature-supported industrial regimes and common event types.
>
> Reference: [1] SMT2020-A Semiconductor Manufacturing Testbed: IEEE Trans. Semicond. Manuf
> > SSI sensitivity to weighting/normalization.
>
> The SSI stability is evaluated under perturbations. Across 2829 runs, the default weighting (0.25, 0.25, 0.25, 0.25) and four extreme schemes (one weight 0.55, others 0.15) yield highly consistent rankings (Spearman $\rho\in[0.91,0.99]$, Kendall $\tau\in[0.76,0.98]$). An 8-step normalizer sweep exhibits minimum $\rho\geq 0.90$ and $\tau\geq 0.76$. Replacing the pipeline with Min-Max, Rank-Quantile, or Z-score yields $\rho\geq 0.836$ and $\tau\geq 0.666$, with a $1.143$--$1.171$ mean absolute decile shift and bounded downstream comparisons.
>
> Thus, SSI preserves a stable macro-level difficulty ordering under both weighting and normalization perturbations.
> > Coverage of DynaSched-Subset.
>
> The DynaSched-Subset (70 instances: 30 Grid, 40 Sweep) is extracted via greedy $k$-center sampling over 6 normalized metrics, seeded by the maximum-difficulty instance, preserving the stress spectrum. Ranges are difficulty $[4.58,30.47]$ vs. $[4.58,33.12]$(full); utilization $[0.30,0.96]$ vs. $[0.18,1.43]$; and arrival-SCV $[0.23,9.02]$ vs. $[0.18,17.16]$, with $100\\%$ IQR coverage and KS $0.175/0.086/0.251$. The [difficulty(click)](https://anonymous.4open.science/r/DynaSchedBench-icml-rebuttal-3551/combined_difficulty_ecdf_full_vs_subset.png) and [driver-metric(click)](https://anonymous.4open.science/r/DynaSchedBench-icml-rebuttal-3551/combined_metrics_ecdf_full_vs_subset.png) ECDF plots capture the broader distributional shapes.
>
> These results confirm that the subset faithfully preserves the full difficulty and metric distributions rather than only matching extrema.
> > Distributional evidence and robustness.
>
> Mean $C_{\max}$ is not the only signal: our method isolates the Observability Paradox by worsening the tails:
> |Config.|Mean|p50|IQR|p90|p95|
> |---|---:|---:|---:|---:|---:|
> |L1 Direct|3.01%|0.65%|1.69%|3.56%|9.75%|
> |L1+Tool|3.08%|0.61%|1.47%|4.67%|11.85%|
> |L2 CoT|1.71%|0.27%|1.04%|2.12%|6.66%|
> |L3 CoT|2.72%|0.39%|1.04%|3.12%|8.91%|
>
> This shows that richer structural priors primarily harm worst-case stability rather than average performance, which is precisely the pattern underlying the Observability Paradox. Against heuristics, Qwen3-8B exhibits $1.01/0.53/0.85/2.85/3.25\\%$ (mean/p50/IQR/p90/p95), whereas the best heuristic achieves $1.11/0.97/1.40/2.43/2.69\\%$.
>
> Therefore, both the tail-side Observability-Paradox pattern and the heuristic-level ceiling remain stable under subset perturbations.
> > SESC operator contributions.
>
> To measure the impact of each contribution on the final accuracy, we test SESC while removing each feature at a time across 111 configurations with 5 replicates (555 runs/variant), reporting the normalized $L_2$ error degradation $\Delta$. Slack scaling dominates ($+0.0225$ mean, $+0.0902$ p90). Removing bottleneck engineering yields small impacts ($+0.0007$, $+0.0056$ p90). Removing processing-time resampling or arrival structure alters the mean ($-0.0077$, $-0.0124$) but worsens the tails ($+0.0011$, $+0.0078$ p90). The minimal subsets recover average-case accuracy ($+0.0036$ to $+0.0122$ mean $\Delta$) but incur tail degradation ($+0.0528$ to $+0.0603$ p90).
>
> These results show that no reduced subset can replace full SESC uniformly: different operators dominate different difficulty regions, and the full operator set is needed for reliable calibration across both average and high-stress cases.
> > Applicability beyond step-wise dispatching.
>
> Step-wise evaluation exposes limitations of reactive LLM dispatching (high-dimensional observations, unstable long-tail behavior), motivating hybrid/offline LLM roles. In our code-as-policy study, the shop state is compressed into sliding-window statistics. The LLM generates Python code allocating numerical priority scores, circumventing token-by-token online selection. This decouples expensive reasoning from online execution, mitigating API latency and the Observability Paradox. DynaSchedBench naturally supports hybrid LLM schedulers (policy-code generation, rule discovery, DRL guidance), evaluating generalization across controlled difficulty axes rather than overfitting static instances.

---

### Decision · Program_Chairs · 2026-04-30

**Decision:**

Accept (regular)

**Comment:**

The reviewers agreed that this paper makes a useful contribution in the form of a calibrated benchmark framework for dynamic scheduling. They found strengths in the event-space calibration idea, the modular and reproducible simulation stack, and the empirical study of LLM-based schedulers. The rebuttal addressed several important concerns by adding evidence on SSI sensitivity, subset coverage, operator ablations, structural validity, and controlled analyses of the L2/L3 prompt difference.

Reviewers still differed on several points, especially the novelty and generality of SESC beyond the DFJSP setting, the fairness of the comparison with parameter-space MOO/Hybrid baselines, and whether SSI and the “Observability Paradox” are validated strongly enough for the paper’s broadest claims. These concerns limit the paper’s impact and generality. However, the reviewers generally agreed that the paper is technically sound, reasonably well supported empirically, and likely to be useful to part of the ICML community as a benchmark and evaluation resource.